# Epigenetic modulation of immune synaptic-cytoskeletal networks potentiates γδ T cell-mediated cytotoxicity in lung cancer

Rueyhung R. Weng [1], Hsuan-Hsuan Lu [1], Chien-Ting Lin[2,3], Chia-Chi Fan[4], Rong-Shan Lin[2,3], Tai-Chung Huang [1], Shu-Yung Lin [1], Yi-Jhen Huang [4], Yi-Hsiu Juan[1], Yi-Chieh Wu[4], Zheng-Ci Hung[1], Chi Liu[5], Xuan-Hui Lin[2,3], Wan-Chen Hsieh [6,7], Tzu-Yuan Chiu[8], Jung-Chi Liao [8], Yen-Ling Chiu[9,10,11], Shih-Yu Chen[6], Chong-Jen Yu [1,12] & Hsing-Chen Tsai [1,4,11✉]

γδ T cells are a distinct subgroup of T cells that bridge the innate and adaptive immune system and can attack cancer cells in an MHC-unrestricted manner. Trials of adoptive γδ T cell transfer in solid tumors have had limited success. Here, we show that DNA methyltransferase inhibitors (DNMTis) upregulate surface molecules on cancer cells related to γδ T cell activation using quantitative surface proteomics. DNMTi treatment of human lung cancer potentiates tumor lysis by ex vivo-expanded Vδ1-enriched γδ T cells. Mechanistically, DNMTi enhances immune synapse formation and mediates cytoskeletal reorganization via coordinated alterations of DNA methylation and chromatin accessibility. Genetic depletion of adhesion molecules or pharmacological inhibition of actin polymerization abolishes the potentiating effect of DNMTi. Clinically, the DNMTi-associated cytoskeleton signature stratifies lung cancer patients prognostically. These results support a combinatorial strategy of DNMTis and γδ T cell-based immunotherapy in lung cancer management.

[1] Department of Internal Medicine, National Taiwan University Hospital, Taipei, Taiwan. [2] Tai Cheng Stem Cell Therapy Center, National Taiwan University, Taipei, Taiwan. [3] Pell Biomedical Technology Ltd, Taipei, Taiwan. [4] Graduate Institute of Toxicology, College of Medicine, National Taiwan University, Taipei, Taiwan. [5] Department of Plant Pathology and Microbiology, National Taiwan University, Taipei, Taiwan. [6] Institute of Biomedical Sciences, Academia Sinica, Taipei, Taiwan. [7] Genome and Systems Biology Degree Program, National Taiwan University, Taipei, Taiwan. [8] Institute of Atomic and Molecular Sciences, Academia Sinica, Taipei, Taiwan. [9] Graduate Program in Biomedical Informatics, Department of Computer Science and Engineering, College of Informatics, Yuan Ze University, Taoyuan, Taiwan. [10] Department of Medical Research, Far Eastern Memorial Hospital, New Taipei City, Taiwan. [11] Graduate Institute of Clinical Medicine, College of Medicine, National Taiwan University, Taipei, Taiwan. [12] Department of Internal Medicine, College of Medicine, National Taiwan University, Taipei, Taiwan. ✉email: htsai@ntu.edu.tw

mmune-based therapy has become a promising strategy in the treatment of lung cancer. Nevertheless, only a small subset of patients respond to current checkpoint blockade therapies. Therefore, the adoptive transfer of immune cells targeting tumor cells has emerged as an attractive therapeutic option[1–3]. γδ T cells, a distinct subset of T cells that sense molecular stress signals from pathogen-infected or transformed cells, can display broad tumor-targeting capabilities in a major histocompatibility complex (MHC)-independent manner[4]. These cells provide potential therapeutic opportunities for patients who are inherently unresponsive to checkpoint blockade due to low mutational burden, or have acquired defects along the TCR-MHC axis. However, the adoptive transfer of autologous or allogeneic γδ T cells in earlier clinical trials showed limited efficacy[5–7]. Thus, a better understanding of tumor lytic interactions between γδ T and cancer cells are required to maximize the therapeutic efficacy of γδ T-based immunotherapy.

DNA methyltransferase inhibitors (DNMTis) have been approved by the U.S. FDA for the treatment of hematological malignancies. Nevertheless, the therapeutic efficacies of these drugs used as a single agent in solid tumors have been less than satisfactory. Compelling evidence has indicated the immunomodulatory effects of DNMTis that target MHC and T cell receptor (TCR) axis[8–12]. On the other hand, little is known about whether and how DNMTis may reshape surface proteins of cancer cells to facilitate MHC-unrestricted recognition and tumor lysis in cell-based immunotherapies. Moreover, molecular characteristics of patients that may potentially benefit from this kind of immunomodulation have not been identified.

Here we show that DNMTis markedly alter the surface proteome of lung cancer cells using stable isotope labeling by amino acids in cell culture (SILAC)-based quantitative proteomics. These experiments show upregulation of multiple MHC-unrestricted immune recognition molecules following transient DNMTi treatment at low doses. Gene ontology analysis of the DNMTi-induced surface proteome reveals a high association with γδ T cell activation. We demonstrate that DNMTi treatment of cancer cells enhances immune synapse formation between ex vivo expanded allogeneic γδ T cells and cancer cells and potentiates antitumor immunity by γδ T cells. A combined genome-wide analysis of the DNA methylome, mRNA-seq, and Omni-ATAC-seq reveals coordinated epigenetic regulatory patterns of immune synaptic cytoskeleton networks. Additionally, primary lung cancer tissues can be stratified by immune cytoskeleton patterns indicative of responses to γδ T-mediated cytotoxicity, which may aid in patient selection in future clinical trials for precision immunotherapy. These findings support the broad applicability of DNMTis in remodeling the cancer cell cytoskeleton for MHC-unrestricted γδ T cell-based therapy.

## Results

**DNMTi upregulates surface molecules related to γδ T cell activation.** Two FDA-approved DNMTis, decitabine (Dacogen, DAC), and azacytidine (Vidaza, AZA) are cytidine analogs that incorporate into DNA/RNA and deplete DNMTs through irreversible binding of the enzymes followed by proteasome degradation[13]. DNMTis have been shown to modulate pathways related to antigen processing/presentation, human leukocyte antigens (HLA), and interferon responses across different cancer types at the transcriptomic level[11,12].

Nevertheless, these transcriptomic changes may not fully reflect the alterations of surface proteins that account for susceptibility to immunotherapy. Thus, we sought to obtain a comprehensive profile of surface proteins altered by decitabine (DAC) through

isolating cell surface proteins from A549 human lung cancer cells before and after DAC treatment using the EZ-link Sulfo-NHS-SS-biotin-assisted biotinylation method, followed by a SILAC-based quantitative proteomics approach (Fig. 1a)[14]. We employed a low-dose treatment protocol established in our previous study to manifest the drug's epigenetic effects over cytotoxicity[15]. First, lung cancer cells were differentially labeled by growing in cell culture media containing heavy arginine (L- Arginine-$^{13}C_6$)/lysine (L-Lysine-$^{13}C_6$,) or normal arginine/lysine. The SILAC-labeled cells (heavy) and the unlabeled cells (light) were treated with phosphate-buffered saline and 100 nM DAC, respectively, for three consecutive days (D3) and grown in a drug-free medium for another three days (D3R3) (Fig. 1b). After biotinylation of the cell surface proteins, total cell lysates of DAC-treated cells at D3 and D3R3 were 1:1 mixed with their corresponding mock-treated counterparts and subjected to surface protein isolation followed by quantitative proteomics. We identified 666 and 831 Gene Ontology (GO)-annotated surface proteins (corresponding to 8791 and 11,898 unique peptides) in A549 cells at D3 and D3R3, respectively. (Fig. 1c and Supplementary Fig. 1a). The continued increase in identified surface proteins after drug withdrawal is consistent with our prior report of epigenetic memory effects following transient drug exposure[15]. Among all the identified proteins, we focused on 314 proteins that showed sustained upregulation upon DAC treatment by at least 1.4-fold for both D3 and D3R3, for their continued induction after drug withdrawal suggested a possible epigenetic regulation (Fig. 1d). Many of these are immune-related proteins. In addition to MHC molecules and certain immune checkpoint proteins that were previously known to be upregulated by DAC[11,12], we uncovered a plethora of proteins that participate in innate immunity or MHC-unrestricted immunity, such as MHC class I polypeptide-related sequences A and B (i.e., MICA, MICB), which are ligands of NKG2D on γδ T and NK cells (Fig. 1e)[16,17]. DAC also upregulates UL16-binding proteins 2 and 3 (i.e., ULBP2 and ULBP3), another group of NKG2D ligands that are expressed in many cancers and in stressed/damaged tissues (Fig. 1e)[18–20]. These molecules have been identified as targets for tumor immunosurveillance by the innate immune system and may elicit antitumor immunity without the requirement for conventional MHC-restricted antigen presentation[21]. In addition, the two death receptors (i.e., TRAIL-R1 and TRAIL-R2) for TNF-related apoptosis-inducing ligand (TRAIL) that induce cancer apoptosis as part of immune surveillance[22] were significantly upregulated at D3R3 (Fig. 1e).

Interestingly, analysis of SILAC-based surfaceomes in another two human lung cancer cell lines, H1299 and CL1-0, revealed highly similar profiles of DAC-induced surface proteins. There were 431 proteins commonly upregulated by DAC in all three lung cancer cell lines for D3R3 (Fig. 1f and Supplementary Fig. 1b). The involvement of DAC-induced surface molecules in the innate immune response is evident, and we mapped these proteins against the innate immune interactomes from InnateDB, an extensively curated database of innate immune pathways and interactions[23]. We uncovered DAC-mediated innate immune molecules that participate in adhesion/cell-cell interactions (e.g., CD97 and ICAM-1), the cytoskeleton (e.g., ARPC2 and VASP), heat shock protein responses, integrin-associated pathways, and various signal transduction networks (Fig. 1g). Next, we performed Gene Ontology (GO) term enrichment using PANTHER Gene List Analysis tools[24] on the DAC-induced surface proteome to search for relevant immune pathways. Remarkably, γδ T cell activation was the top enriched pathway among all immune-related processes, followed by natural killer (NK) cell-mediated immunity — two of the key immune cell types involved in innate immune responses against cancer (Fig. 1h)[7,25].

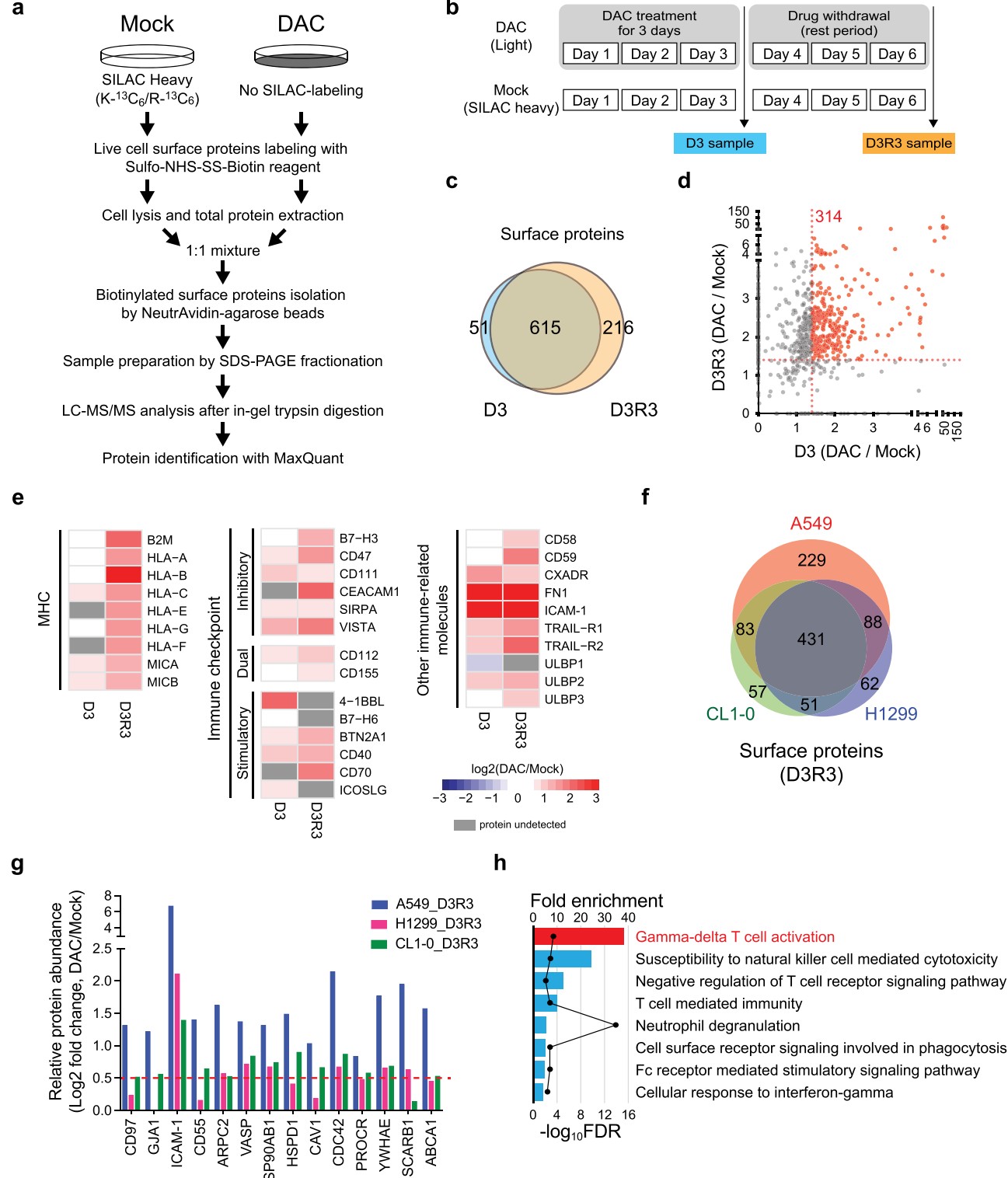

**Fig. 1 Decitabine upregulates surface immune molecules related to γδ T cell activation. a** Experimental diagram of stable isotope labeling with amino acids in cell culture (SILAC)-based quantitative proteomics on biotinylated surface proteins in mock-treated vs. decitabine (DAC)-treated lung cancer cells. SDS-PAGE sodium dodecyl sulfate polyacrylamide gel electrophoresis, LC-MS/MS liquid chromatography-tandem mass spectrometry. **b** Treatment schedule of DAC at 100 nM daily for 72 h (D3), followed by drug withdrawal for 3 days (D3R3). **c** Venn diagram showing the numbers of surface proteins identified at D3 and D3R3 in A549 cells. **d** A scatter plot of proteins upregulated at D3 and D3R3 in A549 cells following decitabine treatment. **e** Heatmap showing log2 fold changes of immune-related surface molecules in DAC-treated vs. mock-treated A549 cells at D3 and D3R3. **f** Venn diagram showing numbers of surface proteins commonly identified at D3R3 in A549, H1299, and CL1-0 cells. **g** Bar graphs showing relative protein abundance of selected surface proteins related to innate immunity in surface proteomes of A549, H1299, and CL1-0 cells following decitabine treatment at D3R3 as compare with mock-treated cells. $n = 1$ for each treatment of individual cell lines. **h** PANTHER gene list analysis on immune-related pathways for proteins upregulated by decitabine at D3R3. FDR false discovery rate.

**Ex vivo expansion of human Vδ1 γδ T cells with antitumor functions.** Based on the DAC-mediated surfaceome data, we reason that epigenetic therapy has the potential to enhance tumor attack by γδ T cells. To conduct preclinical functional studies with γδ T cells for their therapeutic potential, our group further optimized a sequential cytokine stimulation protocol in the presence of γδ TCR/CD3 agonist antibodies for ex vivo clinical-grade expansion of γδ T cells preferentially enriched for the Vδ1+ subset, "Delta One T" (DOT cells)[26], from the peripheral blood of healthy donors. The process did not require the step of initial γδ T cell purification and αβ T cell depletion shown in prior reports[26–28]. Over the course of 21 days, the percentage of total γδ T cells in the CD3+ cell population increased from less than 10% to over 90%, while Vδ1+ T cells accounted for approximately 70% of all γδ T cells (Fig. 2a and Supplementary Fig. 2a). Consistently, we observed a 1571–14,245-fold expansion of Vδ1 T cells (Day 21/Day 0), but only 3 to 59-fold for Vδ2 T cells in another five healthy donors using the same protocol (Supplementary Fig. 2b, c). To evaluate the immunophenotypes of expanded γδ T cells, we performed single-cell analysis on PBMCs at baseline and after ex vivo expansion using mass cytometry with an antibody panel designed to characterize T cell and MHC-unrestricted immunity-related receptors. We used t-distributed stochastic neighbor embedding (t-SNE) to investigate the differences and cellular heterogeneity within T cells before and after ex vivo expansion. Preferential enrichment of the Vδ1+ subset from the peripheral blood after 14 days of expansion was noted (Fig. 2b and Supplementary Fig. 3a, b). Within the expanded Vδ1+ subset, we observed differential expression of MHC-unrestricted immunity-related receptors, including NKG2D, CD226, CD244, NTB, CRACC, and NKp30 and the expression levels of these receptors were highly correlated (Fig. 2c). Furthermore, our ex vivo expanded γδ T cells express markers for cytolytic degranulation (e.g., CD107a) as well as secrete antitumor effector cytokines (i.e., TNF and IFN-γ). On the other hand, the expanded γδ T cells secrete almost no protumor or negative regulatory cytokines, such as IL-17A and IL-10, nor do they express PD-1, an immune checkpoint associated with the exhausted state (Fig. 2b and Supplementary Fig. 3b, c). which precludes the concern that constant cytokine stimulation in the ex vivo expansion protocol leads to premature exhaustion of γδ T cells. The data indicate that the expanded DOT-like cells (referred to as γδ T cells below for simplicity) are at an activated and proliferative stage with the antitumor phenotype suitable for potential therapeutic uses.

**DNMTi enhances γδ T cell-mediated cytolysis of lung cancer cells.** Subsequently, we investigated whether pretreatment with low-dose DAC may potentiate the susceptibility of lung cancer cells to the attack by γδ T cells. First, we found that in vitro coculture of untreated lung cancer and γδ T cells at an effector to target (E:T) ratio of 3:1 caused merely 20–30% cytolysis of A549 human lung cancer cells (Supplementary Fig. 4a). Remarkably, 72-h daily treatment with 100 nM DAC on human lung cancer cells (i.e., HCC827, H1299, and A549 cells) significantly potentiates the killing of lung cancer cells by γδ T cells at the same E:T ratio. On the other hand, either treatment alone resulted in minimal or moderate cell death using annexin V and propidium iodide apoptosis assays (Fig. 2d, e). We observed similar potentiating effects in several other lung cancer cell lines, including H2981, PC9, PC9-IR (Iressa resistant), H157, CL1-0, and CL1-5, as well as patient-derived lung cancer cells from a malignant pleural effusion, PDC#0899 (Fig. 2f and Supplementary Fig. 4b). We also tracked DAC's potentiation effect on γδ T cell killing using an electric cell-substrate impedance sensing

(ECIS) system, a biophysical approach for real-time monitoring of the γδ T cell killing process. γδ T cell-mediated cytolysis usually takes place within 30 min to a few hours. The effect can be enhanced by DAC pretreatment at a dose that does not cause significant cytotoxicity (Supplementary Fig. 4c). Notably, this potentiating effect can be observed in other tumor types, such as HCT116 colon cancer cells (Supplementary Fig. 4d). On the other hand, we observed no significant increase in cytotoxicity of PBMCs from healthy donors upon combination treatment of DAC and ex vivo expanded γδ T cells; Supplementary Fig. 4e), which suggested that this potentiation effect may spare normal cells.

In addition to the granule exocytosis-dependent cytotoxic pathway through direct contact, activated γδ T cells may also trigger cancer death via non-contact cytotoxicity by secreting TRAILs that engage TRAIL receptors and the downstream apoptotic pathway[29,30]. To assess the significance of secretory TRAIL-mediated apoptosis in the DAC potentiating effect, we used a transwell system that allows for the diffusion of γδ T cell-secreted TRAILs to reach cancer cells without direct cell contact. No significant cancer cell death was observed after 24 h (Supplementary Fig. 5), suggesting that direct cell contact is required for DAC-potentiated γδ T cell killing. Furthermore, we evaluated whether DAC enhances γδ T cell chemotaxis using a transwell coculture system that allows γδ T cells to pass through the membrane, and there was no significant difference between the numbers of γδ T cells migrating toward mock-treated or DAC-treated lung cancer cells in the bottom wells. The data suggest that the potentiating effect of DAC is not dependent on increased chemoattraction of γδ T cells by DAC-primed cancer cells (Fig. 2g).

**DNMTi facilitates synapse formation between cancer and γδ T cells.** Since effective lysis of cancer cells by immune cells relies on functional immune synapses to facilitate directional and coordinated delivery of lytic granules[31], we investigated the efficiency of immune synapse formation between γδ T cells and DAC-pretreated cancer cells by immunofluorescence staining of phosphotyrosine (pTyr), a marker of immune synapses with active signaling[32]. We observed a much higher number of immune synapses formed between γδ T and DAC-treated H1299 lung cancer cells than by mock-treated cells (Fig. 3a). In search of the key molecules involved in the synaptic interaction, we compared the DAC-induced surface proteomes of two human lung cancer cell lines — H1299 and A549, both of which showed enhanced γδ T cell killing following DAC treatment. Among the identified proteins, intercellular adhesion molecule 1 (ICAM-1) was highly upregulated in both cell lines (Fig. 3b). Western blotting confirmed that DAC can significantly upregulate ICAM-1 proteins in many lung cancer cell lines as well as a patient-derived lung cancer cell line from a malignant pleural effusion, PDC#062 (Fig. 3c). ICAM-1 is a surface glycoprotein and a member of the immunoglobulin superfamily. It is present in immune cells, endothelial cells, and epithelial cells as a general adhesion molecule. We demonstrated that ICAM-1 is localized within the cancer-γδ T immune synapse along with other classical immune synapse proteins, including LFA-1 and linker for activation of T cells (LAT; Fig. 3d). Interestingly, knockout of *ICAM1* (KO-*ICAM1*) with CRISPR technology (Supplementary Fig. 6a, b) in lung cancer cells (i.e., A549, H1299, and CL1-0) ultimately diminished the potentiation effects of DAC for γδ T cell killing in all three cell lines tested (Fig. 3e, f). On the other hand, overexpression of *ICAM1* (OV-*ICAM1*) in lung cancer cell lines with a Tet-On system (Supplementary Fig. 6b) markedly enhanced γδ T cell-mediated cytotoxicity on human lung cancer cells,

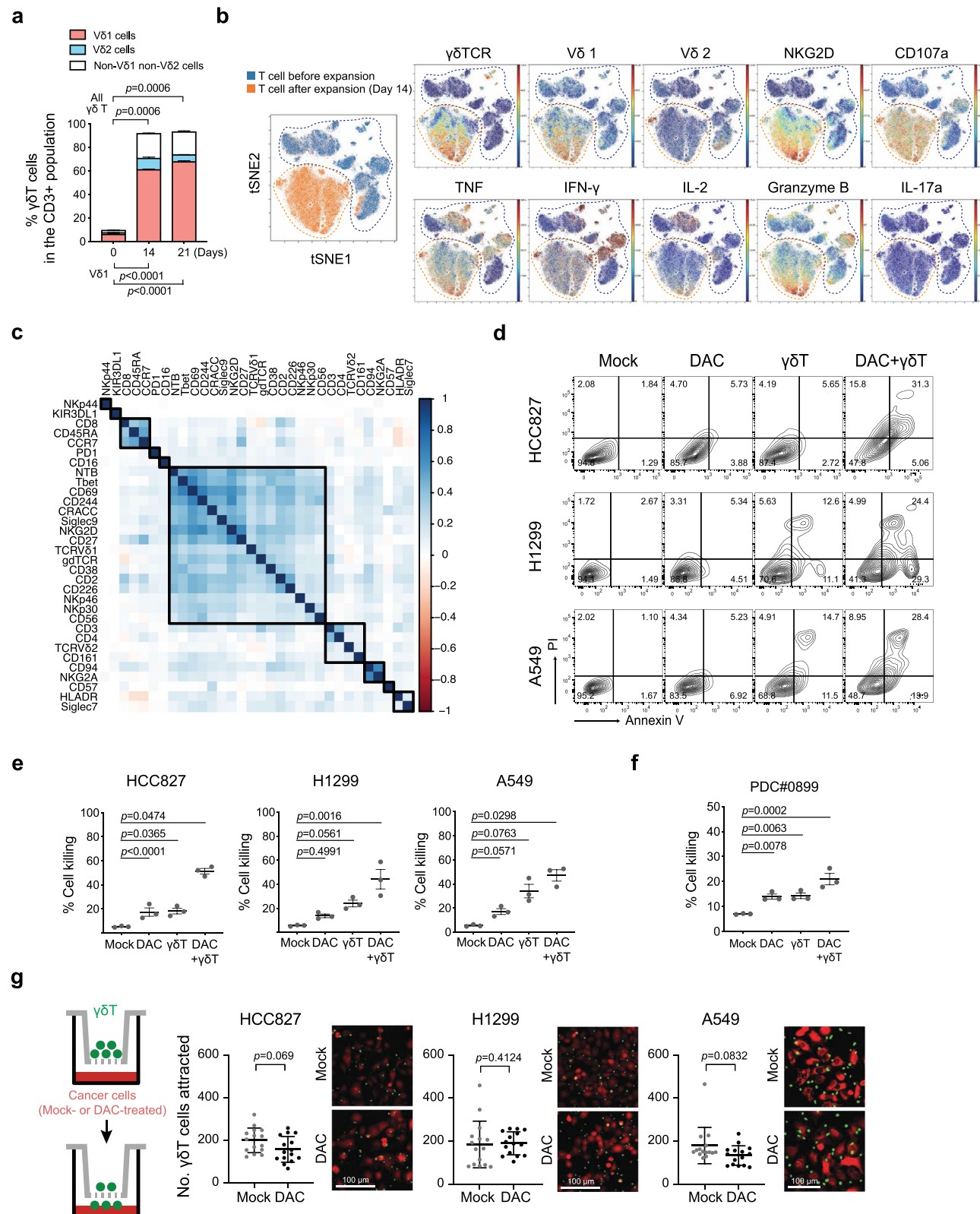

mimicking the effect of DAC pretreatment (Fig. 3g, h). As anticipated, DAC treatment of *ICAM1*-knockout cells failed to enhance immune synapse formation between lung cancer and γδ T cells (Fig. 3i). Collectively, the data suggest that ICAM-1-mediated adhesion and stabilization of the synaptic structure is critical for DAC's potentiating effects on γδ T cell antitumor immunity.

While Vδ1 cells are the major cell subset in the ex vivo-expanded γδ T cells, the cell preparations may contain minor subsets of Vδ2 cells and CD8 T cells (Fig. 2a and Supplementary Fig. 2). To further dissect the potential contributions of each cell subset to the immune synapse formation and tumor lytic activity, we isolated individual cell subsets (i.e., Vδ1, Vδ2, and CD8 T cells) by fluorescence-activated cell sorting (FACS) to perform coculture

**Fig. 2 Decitabine enhances γδ T cell-mediated cytolysis of lung cancer cells. a** Bar graphs showing percentages of γδ T cell subsets in the CD3+ population from the peripheral blood of a healthy donor following ex vivo expansion. Data are presented as mean ± standard error of the mean (SEM). $n = 3$, independent experiments. **b** Overlay of tSNE maps of CD3+ T cells from PBMC at baseline and after ex vivo expansion analyzed by mass cytometry. Each dot represents a single cell. The color represents the expression level of the indicated markers. Red is high, and blue is low. **c** Pearson correlation of markers expressed in ex vivo expanded γδ T cells analyzed by mass cytometry. **d** Representative flow cytometric analysis of annexin V and propidium iodide (PI) apoptosis assays in human lung cancer cell lines upon treatments with 100 nM decitabine (DAC) alone, γδ T cells alone or DAC/γδ T cell combination. The effector to target (E:T) ratio is 3:1. Gating strategies are shown in Supplementary Fig. 16. **e** Dot plots showing three biological replicates (mean ± SEM) of apoptosis assays described in **d**. **f** Annexin V and propidium iodide apoptosis assays of a patient-derived lung cancer cell line from malignant pleural effusion, PD#0899 (mean ± SEM, $n = 3$, independent experiments). **g** Transwell migration assays of γδ T cells (upper chamber) towards mock- or DAC-treated lung cancer cells (lower chamber). Numbers of γδ T cells in the lower chambers are counted per high power field. Representative images of γδ T cells (Hoechst 33342-labeled; green) and lung cancer cells (Calcein-retained; red) in the lower chambers are shown. Data are presented as mean ± standard deviation (SD). $n = 15$ high power fields, over three independent experiments. Scale bar: 100 μm. The $p$ value is calculated by one-way ANOVA with Tukey's multiple comparison test (panels **a**, **e** and **f**) or the two-sided Mann–Whitney test (**g**).

experiments with H1299 lung cancer cells. Notably, Vδ1 cells expressed higher levels of CD107a and displayed a stronger cytolytic capacity than Vδ2 or CD8 T cells (Supplementary Fig. 7a, b). We also observed a greater number of immune synapses formed between cancer cells and Vδ1 cells than Vδ2 or CD8 T cells (Supplementary Fig. 7c). The data indicate that Vδ1 cells were probably the major player in γδ T cell-mediated antitumor responses as they were dominant in cell numbers as well as functional capacities. Consistently, it is worth noting that the putative ligands for γδ TCRs on non-Vδ1 cells (e.g., Vδ2 cells), including BTN2A1[33,34], BTN3A[35], and others[36], are not consistently upregulated by DAC in all the cancer cell lines from our surface proteomics study (Supplementary Fig. 8a), nor is BTNL3[37], a ligand for Vγ4Vδ1 TCRs. Such ligands therefore may not be essential for γδ T cell cytolytic activity in this context.

In addition, we performed γδ TCR and NKG2D blocking assays to determine whether DAC-mediated ICAM-1 upregulation may override the conventional tumor recognition mechanisms through γδ TCR or NKG2D at the interface between γδ T cells and DAC-pretreated lung cancer cells. Notably, blocking of NKG2D attenuated γδ T-mediated cytolysis of DAC-pretreated lung cancer cells (Supplementary Fig. 8b), which suggested that the NKG2D receptor-ligand axis is also important for immune recognition by γδ T cells and the induction of ICAM-1 by DNMTi further strengthens the cell–cell interaction required for successful tumor lysis.

**DNMTi stabilizes synaptic clefts via rearranging the cytoskeleton.** As we further deciphered how DAC affects the synaptic structure to exert MHC-unrestricted cytotoxicity, we observed a marked accumulation of filamentous actin (F-actin) at the cancer cell membrane near the region of immune synapses (Fig. 4a, b). When we knocked out *ICAM1* in H1299 lung cancer cells, there was a substantial decrease in DAC-induced F-actin accumulation at immune synapses (Fig. 4c and Supplementary Fig. 9a), which was associated with a significant reduction in synaptic cleft width (Fig. 4d). In contrast, DAC-treated lung cancer cells with normal ICAM-1 expression established immune synapses with a prominent synaptic cleft between cancer and γδ T cells (Fig. 4d). Notably, pharmacological disruption of actin polymerization with cytochalasin B (Cyto B) abolished DAC-induced F-actin clustering and pTyr signaling at immune synapses but appeared to have minimal effects on ICAM-1 expression (Fig. 4e and Supplementary Fig. 9b). Additionally, cytochalasin B also diminished the DAC effects on the width of the synaptic cleft (Supplementary Fig. 9c, d). Furthermore, cytochalasin B significantly counteracted DAC's enhancing effects on immune synapse formation (Fig. 4f, g). Since F-actin accumulation and larger sizes of the synaptic clefts are characteristics of activating instead of inhibitory immune

synapses[38], our data suggest that DAC remodels the actin cytoskeleton to facilitate the formation of activating immune synapses between cancer and γδ T cells.

**Depletion of DNMTs induces γδ T-sensitive cytoskeletal gene patterns.** Proper cytoskeleton dynamics and arrangements in immune cells are critical for satisfactory immune responses. Nevertheless, the expression patterns of the cytoskeleton at the cancer cell side of the immune synapse, their regulation, and the associated functional consequences remain incompletely understood. In light of our observation of DAC-induced actin cytoskeleton reorganization (Fig. 4), we speculate that there may be coordinated regulation of immune-related cytoskeletal gene networks by disruption of DNMTs. Indeed, genome-wide mRNA-seq data of five lung cancer cell lines (i.e., A549, CL1-0, CL1-5, PC9, and H1299) after DAC treatment showed a significant induction of genes/enzymes involved in cytoskeletal dynamics and reorganization, including *CORO1A*[39], *HCLS1*[40], *FES*[41], among others (Fig. 5a). Gene set enrichment analysis (GSEA) also revealed a striking enrichment for gene sets related to actin cytoskeleton reorganization as well as intermediate filament-based processes. In contrast, the gene sets related to microtubules appeared to be downregulated (Fig. 5b, c). Likewise, genetic depletion of DNMTs recapitulates similar cytoskeletal gene expression profiles, as we analyzed the transcriptomic data of colon cancer cell lines (e.g., HCT116 and DLD1) subject to shRNA targeting of *DNMTs*[42] (Fig. 5d). Thus, the data indicate that the specific cytoskeletal remodeling pattern is DNMT-dependent.

Furthermore, the epigenetic regulation of these cytoskeletal genes after DAC treatment appears to be a highly coordinated process. When we examined promoter DNA methylation status of the genes in the three cytoskeletal gene modules — actin cytoskeleton, intermediate filaments and microtubules — in human lung cancer cells with Infinium MethylationEPIC Bead-Chips, we found that genes in the actin gene module had a higher promoter DNA methylation at baseline and became demethylated with DAC treatment. In contrast, genes in the microtubule module tended to have low baseline methylation levels, which were minimally altered by DAC (Fig. 5e). We performed Omni-ATAC-seq[43] to investigate how chromatin accessibility is involved in DAC transcriptional regulation of cytoskeletal modules. Generally, the genes with high promoter DNA methylation levels at baseline tended to have inaccessible chromatin, which gained modest but critical accessibility after DAC treatment. Interestingly, genes with low basal promoter DNA methylation that are upregulated by DAC have preexisting accessible chromatin at the TSS, which remains accessible following DAC treatment (Supplementary Fig. 10). When we carefully examined the chromatin patterns at the cytoskeletal-related genes, we observed moderate chromatin

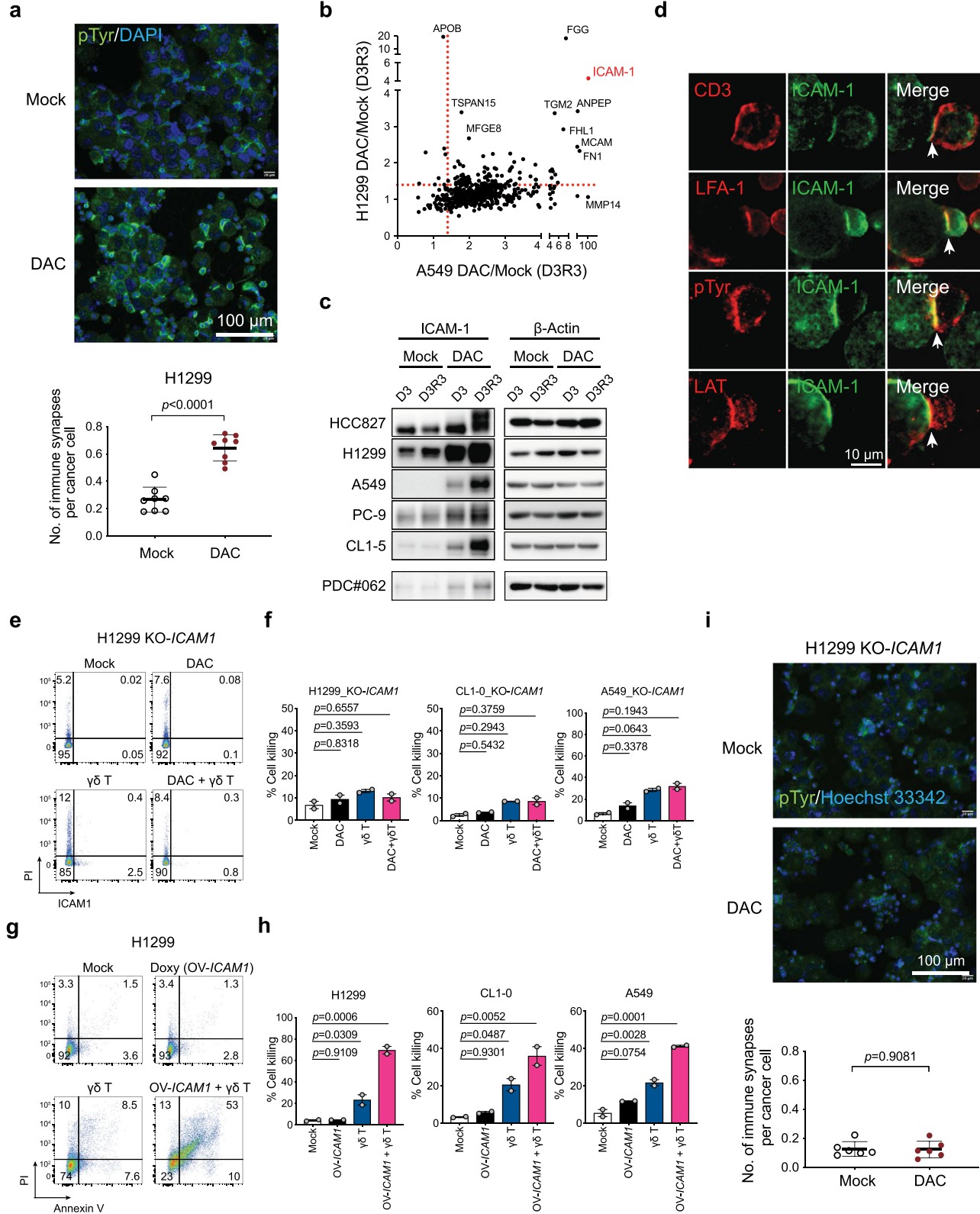

changes at the gene promoters in actin- or intermediate filament-related processes (Fig. 5f). In contrast, there were marked decreases in promoter chromatin accessibilities at the microtubule-related genes downregulated by DAC (Fig. 5f). The coordinated regulatory patterns can be exemplified by individual genes in each of the cytoskeletal modules— *EVPLL*, *TUBE1*, and *DAPK3* (Supplementary Fig. 11).

**TP53 is a hub for the synaptic cytoskeleton and epigenetic proteins**. To further elucidate the relationships between epigenetic proteins and immune synaptic cytoskeleton, we performed a gene network analysis of DAC transcriptomes in lung cancer cell lines by Ingenuity Pathway Analysis (IPA), which revealed an intimate interconnectedness between immune synaptic molecules, the cytoskeleton, and epigenetic proteins (Supplementary

**Fig. 3 Decitabine facilitates immune synapse formation between lung cancer and γδ T cells. a** Immunofluorescence imaging of immune synapses between H1299 lung cancer and γδ T cells by phosphotyrosine (pTyr) staining. Quantifications of immune synapses per cancer cell on eight randomly taken high power fields for each treatment are shown in the dot plots (mean ± SD). Scale bar: 100 μm. DAPI: 4′,6-diamidino-2-phenylindole. **b** A scatter plot of decitabine (DAC)-induced surface proteomes in H1299 and A549 cells at D3R3. **c** Representative western blot analyses of ICAM-1 expression in mock- vs. DAC-treated human lung cancer cells. β-actin: loading control. Two independent experiments were performed. **d** Immunofluorescence staining of ICAM-1 at immune synapses between γδ T and DAC-treated H1299 lung cancer cells. Scale bar: 10 μm. **e** Apoptosis assays of H1299 lung cancer cells with CRISPR-knockout of *ICAM1* (KO-*ICAM1*) subject to γδ T cell killing for 2 h. Lung cancer cells are treated with mock, DAC alone, γδ T cells alone or a combination of DAC and γδ T cells. **f** Bar graphs showing cell death of three human lung cancer cell lines (KO-*ICAM1*) subject to γδ T cell killing for 2 h. Data were summarized from two independent knockout clones for each cell line (mean ± SEM). **g** Apoptosis assays of H1299 lung cancer cells with a Tet-on expression system of *ICAM1* (OV-*ICAM1*) subject to γδ T cell killing for 2 h. **h** Bar graphs showing cell death of human lung cancer cell lines with *ICAM1* overexpression subject to γδ T cell killing for 2 h. Data were summarized from two independent overexpression clones for each cell line (mean ± SEM). **i** Immunofluorescence imaging of immune synapses between H1299 KO-*ICAM1* cells and γδ T cells. Scale bar: 100 μm. Quantifications of immune synapses per cancer cell on six randomly taken high power fields for each treatment are shown in the dot plots (mean ± SD). The *p* value is calculated by the two-sided Mann–Whitney test (**a**, **i**) or one-way ANOVA test (**f**, **h**). Gating strategies for panels **e** and **g** are shown in Supplementary Fig. 16.

Fig. 12a). As shown in the network, we found a general down-regulation of genes involved in microtubule organization (i.e., *TUBG1*, *DCTN1*, and *PLK1*). In contrast, genes participating in actin and intermediate filament dynamics (i.e., *CORO1A*, *GFAP*, and *DES*) were upregulated. *ICAM1* bridges surface immune receptors/HLA molecules to the cytoskeleton in the cytoplasm, which links to *TP53* and other epigenetic modifiers (i.e., *DNMTs*, *HDACs*, and *SMARCA4*) in the nucleus (Supplementary Fig. 12a). Further upstream regulator analysis showed that T cell effector cytokines, such as *IFNG* and *TNF*, may enhance the DAC-induced expression pattern of immune surface molecules, including *ICAM1*, *ICOSLG*, and *HLAs* (Supplementary Fig. 12b). In addition, *TP53* appears to be a hub gene mediating the coordinated changes of immune surface molecules and the cytoskeleton in response to epigenetic modifications, implying that a functional *TP53* network is necessary for effective immune potentiating effects by DAC (Supplementary Fig. 12c).

**DNMTi modulates functional γδ T cell subsets.** To evaluate how DAC may concurrently affect γδ T cells as it would occur in the clinical scenarios, we used mass cytometry to profile both phenotypic and functional immune parameters of expanded γδ T cells with or without DAC treatment. Data from both groups of cells were clustered together by an x-shift algorithm, and the frequency of each subpopulation in untreated and DAC-treated expanded γδ T cells was calculated. As shown in Supplementary Fig. 13a, fourteen clusters within CD3+ T cells were revealed (termed T1 through T14, ranked by the frequency differences between the untreated and DAC-treated groups), each with distinct phenotypic and functional effector signatures. Importantly, the top two cell clusters induced by DAC treatment correspond to Vδ1 and express higher levels of CD226, CD244, CD2, and CRACC together with stronger functional effectors, including CD107A, TNF, granzyme B, IL-2, and IFN-γ, than those expressed by the clusters decreased in cell frequency after DAC treatment (Supplementary Fig 13b). The result was corroborated by Vδ1 T cells expanded from another five healthy individuals treated with 10 nM DAC. We observed a trend of increased production of antitumor effector cytokines, such as IFN-γ and TNF (Supplementary Fig 13c). Moreover, following DAC treatment, there was a marked increase in the percentages of polyfunctional Vδ1+ cells that coexpressed two or more effector cytokines in three of the five individuals (donors #1, #3, and #5; Supplementary Fig. 13d). As polyfunctional T cells are often considered the hallmark of protective immunity[44–46], the data that DAC increases polyfunctionality of γδ T cells further strengthen the rationale of combination therapy using DAC and adoptive transfer of γδ T cells.

**Combined treatment of DNMTi and γδ T cells prolongs survival in mice.** Subsequently, we investigated the in vivo effects of combination therapy with DAC (intraperitoneal injection) followed by the adoptive transfer of ex vivo expanded human γδ T cells (intravenous injection) in immunocompromised NSG (NOD.Cg-Prkdc^scid Il2rg^tm1Wjl/SzJ) mice bearing H1299 human lung cancer xenografts (Fig. 6a). The mice in the combination therapy group had smaller tumors and significantly better overall survival than the mice receiving normal saline or subject to either treatment alone (Fig. 6a, b). Pathologically, the tumor tissues in the combination group were not only smaller but also displayed loose architecture and marked fibrosis. In contrast, tumor tissues in the control group appeared to be hypercellular, with regions of hemorrhage and necrosis (Fig. 6c). Notably, in vivo imaging of γδ T cell trafficking showed enhanced homing ability and significant intratumoral infiltration of γδ T cells in the DAC treatment group of NSG mice bearing lung cancer xenografts a few hours after intravenous injection of γδ T cells, whereas poor intratumoral infiltration of γδ T cells was noted in the control group (Fig. 6d). These data demonstrate the promise of combining DAC with the adoptive transfer of γδ T cells in treating lung cancer in vivo.

**Cytoskeletal gene signatures predict survival of patients with lung cancer.** Proper patient selection is critical for therapeutic success in the design of clinical trials. To investigate if DAC-induced cytoskeletal signatures may potentially stratify patients for their general immunoreactivity and/or for the benefit of epigenetic-primed γδ T cell therapy, we examined the RNA-seq data from the primary lung cancer tissues at National Taiwan University Hospital and the TCGA portal. Unsupervised clustering of primary lung adenocarcinoma tissues based on the core enrichment genes from the actin-, intermediate filament-, and microtubule-related gene modules (Fig. 5c) revealed three groups: immune-sensitive, immune-intermediate, and immune-insensitive (Fig. 7a). Consistently, higher expressions of γδ TCR genes were also detected in the tumor tissues of the immune-sensitive and immune-intermediate groups (Supplementary Fig. 14). To further explore the relationship between the cytoskeletal signature and immune responsiveness, we performed a computational deconvolution of transcriptomic data from the TCGA lung cancer tissues using a CIBERSORT-LM7 reference gene signature matrix, generated by a modified CIBERSORT algorithm[47]. We found that, in addition to γδ T cells, gene signatures for other immune cell types including CD4 and CD8 T cells, B cells, granulocytes, monocytes and NK cells were also enriched in the immune-sensitive group as compared with those in the immune-intermediate and immune-insensitive groups (Supplementary Fig. 15). Remarkably, we observed the best

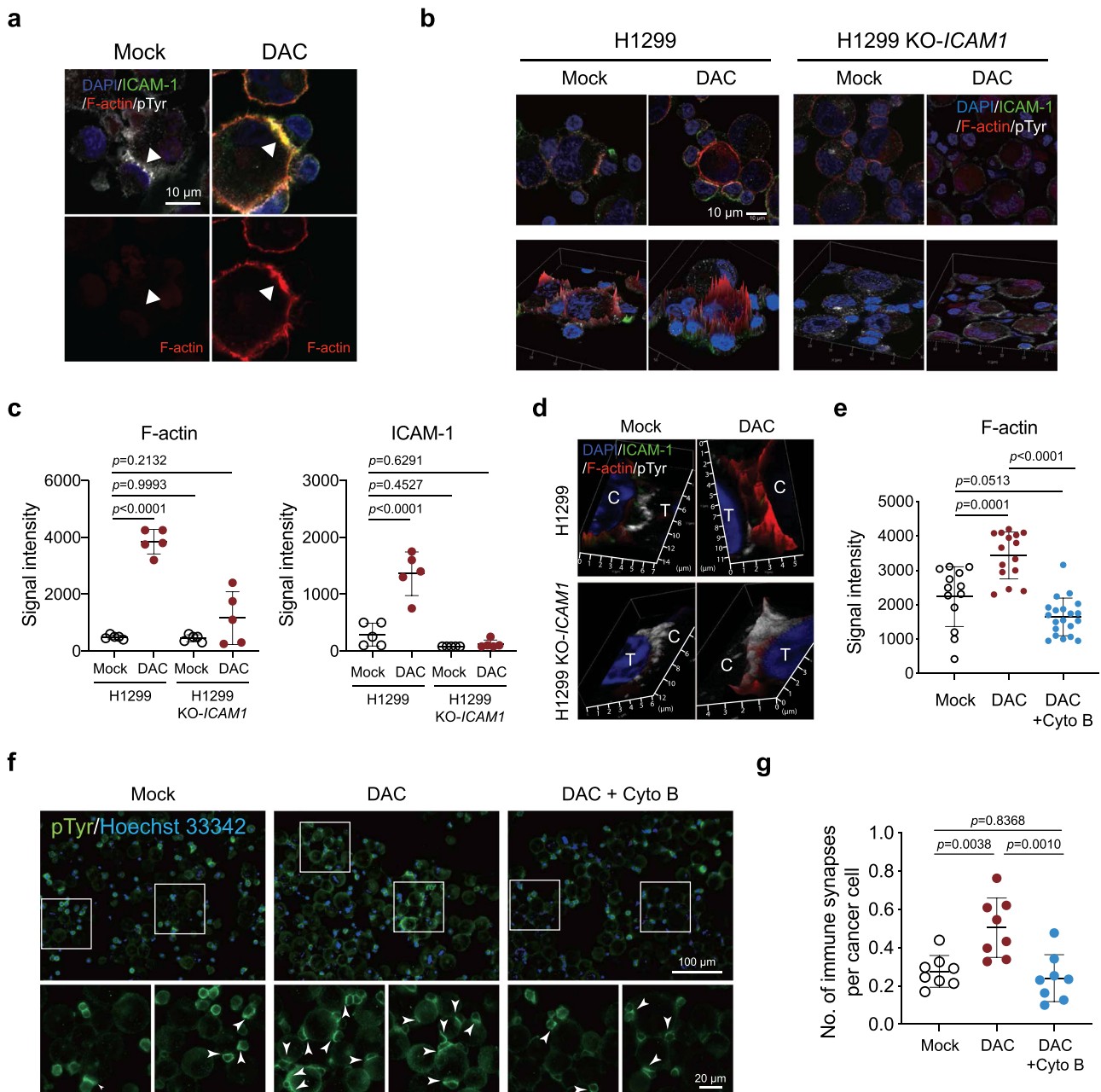

**Fig. 4 Decitabine stabilizes the immune synaptic cleft by strengthening the actin cytoskeleton. a** Immunofluorescence staining of F-actin (red), ICAM-1 (green), pTyr (phosphotyrosine, white) at immune synapses between γδ T cells and DAC-pretreated H1299 lung cancer cells at D3R3. DAPI: 4′,6-diamidino-2-phenylindole. Scale bar: 10 µm. Three independent experiments were performed. **b** Immunofluorescence images of the interfaces between γδ T cells and H1299 lung cancer cells (parental vs. *ICAM1* knockout (KO-*ICAM1*)). Signals of F-actin (red) in the periphery of H1299 cancer cells are shown in two-and-a-half-dimensional (2.5D) images in the lower panels. Scale bar: 10 µm. Three independent experiments were performed. **c** Dot plots of signal intensities of F-actin and ICAM-1 from five pTry-positive immune synapses between γδ T cells and H1299 lung cancer cells (parental or KO-*ICAM1*) from three independent experiments (mean ± SD). **d** Immunofluorescence images of immune synapses between γδ T cells (marked with T) and H1299 lung cancer cells (marked with C) stained for ICAM-1, F-actin, and pTyr. Three independent experiments were performed. **e** Dot plots of F-actin signal intensities at immune synapses between γδ T cells and H1299 cells. H1299 cells are pretreated with PBS (Mock), DAC alone or a combination of DAC pretreatment (D3R3) and 1 µg/mL Cyto B (cytochalasin B) for 1.5 h before coculture with γδ T cells (mean ± SD). n = 13–20 immune synapses over three independent experiments. **f** Immunofluorescence images of immune synapses between γδ T and H1299 cells pretreated with PBS (Mock), DAC alone, and combination of DAC and Cyto B. Blow-up images of the square areas for each treatment are shown in the lower panels. Arrows denote immune synapses between γδ T and H1299 cells. Scale bar: 100 µm (upper) and 20 µm (lower panels). Two independent experiments were performed. **g** Dot plots showing numbers of immune synapses per cancer cell on eight randomly taken high power fields for H1299 cells pretreated with PBS (Mock), DAC, and combination of DAC and Cyto B (mean ± SD). The *p* value is calculated by the two-way ANOVA (**c**) or one-way ANOVA with Tukey's multiple comparisons test (**e, g**).

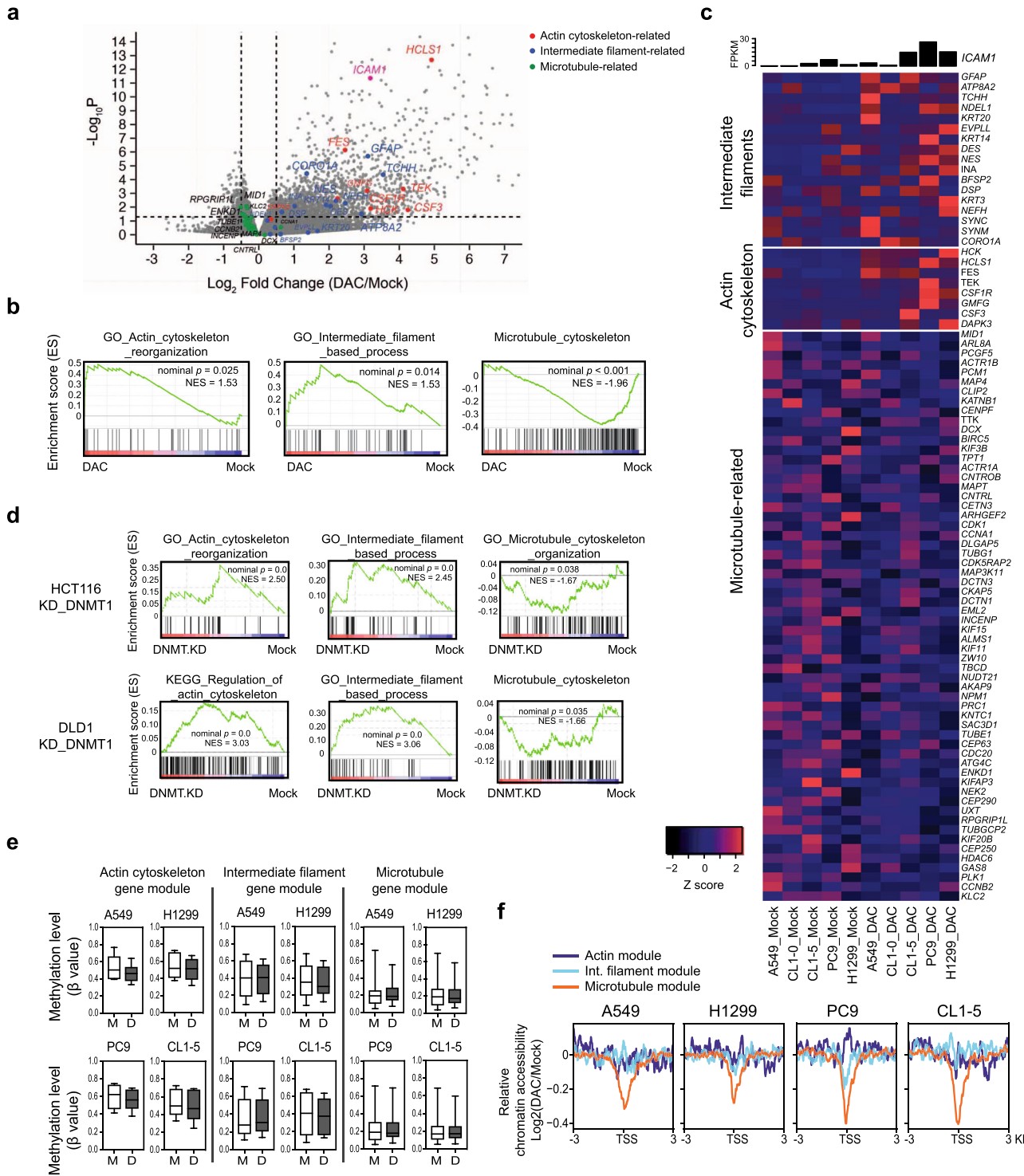

overall survival in the patients with the immune-sensitive signature, corresponding to the DAC-induced immune responsive pattern identified in lung cancer cell lines (the actin and intermediate filament modules upregulated and the microtubule module downregulated; Fig. 7b). In contrast, the immune cell-insensitive signature was associated with the worst prognosis in both NTUH and TCGA cohorts (Fig. 7b). The data highlight the importance of epigenetic reorganization of the cytoskeleton as a clinical indicator for patient outcome and may be used for patient selection to maximize the benefits of γδ T cell-based therapy in lung cancer patients.

## Discussion

DNA methyltransferase inhibitors are versatile immunomodulators[8,48], aside from their antitumor effects[49]. These drugs not only may enhance the immunogenicity of cancer cells[9,12,50,51] but have also been shown to modulate αβ T cell fate decisions, differentiation, and exhaustion[52,53]. Despite the proposed potentiating effects of DNMTis on checkpoint inhibitors[8], the requirement for tumors with high mutational load to trigger conventional αβ TCR-driven, MHC-restricted immune responses limits the added therapeutic benefits of this approach[54]. On the other hand, clinical interest on cell-based immunotherapies,

**Fig. 5 Depletion of DNMTs induces γδ T-sensitive cytoskeletal gene expression in cancer cells. a** A volcano plot showing differentially expressed genes in DAC-treated vs. Mock-treated human lung cancer cells (i.e., A549, CL1-0, CL1-5, PC9, and H1299). The y-axis denotes statistical significance (−log10 of p-value), and the x-axis displays the log2 fold change values between the DAC-treated and the Mock-treated groups. Genes related to the actin cytoskeleton, intermediate filaments, microtubule are marked in red, blue, and green, respectively. **b** Gene Set Enrichment Analysis (GSEA) of mRNA-seq data in five DAC-treated lung cancer cell lines — A549, CL1-0, CL1-5, PC9, and H1299. Gene sets related to actin-cytoskeleton reorganization, intermediate filament-based process, and microtubule-related gene modules are shown. NES Normalized Enrichment Score, GO gene ontology. **c** Heatmap showing mRNA expression of core enrichment genes for actin-cytoskeleton, intermediate filament, and microtubule-related processes in DAC-treated and mock-treated human lung cancer cells measured by mRNA-seq. FPKM fragments per kilobase of transcript per million mapped reads. **d** Gene Set Enrichment Analysis (GSEA) of mRNA-seq data in HCT116 and DLD1 human colorectal cancer cell lines subject to shRNA knockdown of DNA methyltransferase 1 (DNMT1). Gene sets related to actin-cytoskeleton reorganization, intermediate filament-based process, and microtubule-related gene modules are shown. NES Normalized Enrichment Score. **e** Box plots showing promoter methylation status of genes in the actin cytoskeleton-, intermediate filament-, and microtubule-related gene modules in human lung cancer cells treated with 100 nM DAC for three days followed by a 3-day drug-free culture (D3R3). The box denotes the 25th percentile, the median, and the 75th percentile. The whiskers indicate minimum and maximum values. Methylation data are analyzed by Infinium MethylationEPIC arrays. n = 1 for each treatment of individual cell lines. M mock-treated, D DAC-treated. **f** Relative chromatin accessibility around TSS of genes (−3 to +3 kb) in the actin-cytoskeleton reorganization, intermediate filament-based process, and microtubule-related modules in DAC-treated vs. mock-treated lung cancer cell lines. TSS transcription start site.

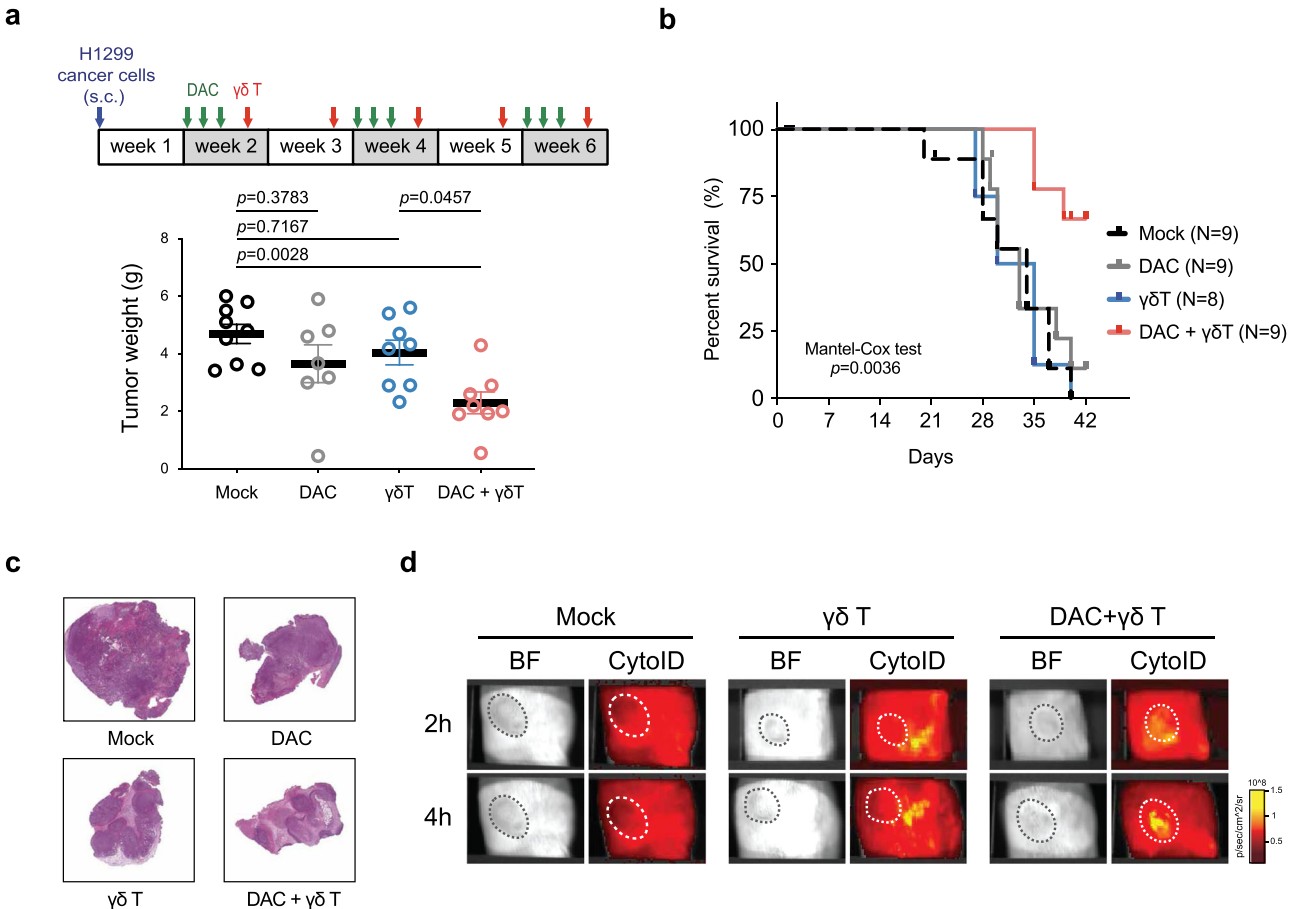

**Fig. 6 Decitabine combined with adoptive transfer of γδ T cells prolongs survival of mice with lung cancer xenografts. a** In vivo experiment of NOD-scid IL2rg^null (NSG) mice bearing H1299 human lung cancer xenografts treated by DAC, adoptive human γδ T transfer, or both. For every cycle of drug treatment, DAC is administered intraperitoneally for three consecutive days, followed by intravenous injection of ex vivo expanded γδ T cells on day 5 and day 12. Mice were killed one day post injection of γδ T cells following the 3rd cycle of the treatment. The tumor was surgically excised, and the weight was measured. The statistic result was determined by one-way ANOVA with Tukey's multiple comparison test. n = 9 mice from two independent experiments (mean ± SEM). s.c. subcutaneously. **b** Kaplan–Meier survival curve of NSG mice in each treatment group is shown. The p value is calculated by the Mantel-Cox test (one-sided). **c** Hematoxylin and eosin (H&E) staining of representative mouse tumors in each treatment group. Images of the whole tumor are generated by a digital slide scanner. **d** In vivo imaging of γδ T cell trafficking in a lung cancer xenograft mouse model. Ex vivo expanded γδ T cells were prestained with CYTO-ID Red long-term cell tracer dye. The NSG mice bearing H1299 lung cancer xenografts were treated with DAC (0.2 mg/Kg BW) or normal saline intraperitoneally for three consecutive days (day 1–3), followed by tail vein injection of the prestained γδ T cells on day 5 and day 12. The images were taken at 2 and 4 h after γδ T injection using IVIS Spectrum (Ex570, Em640) on day 12. Experiments were done in replicates. Representative images are shown here. BF bright field, White circle tumor region.

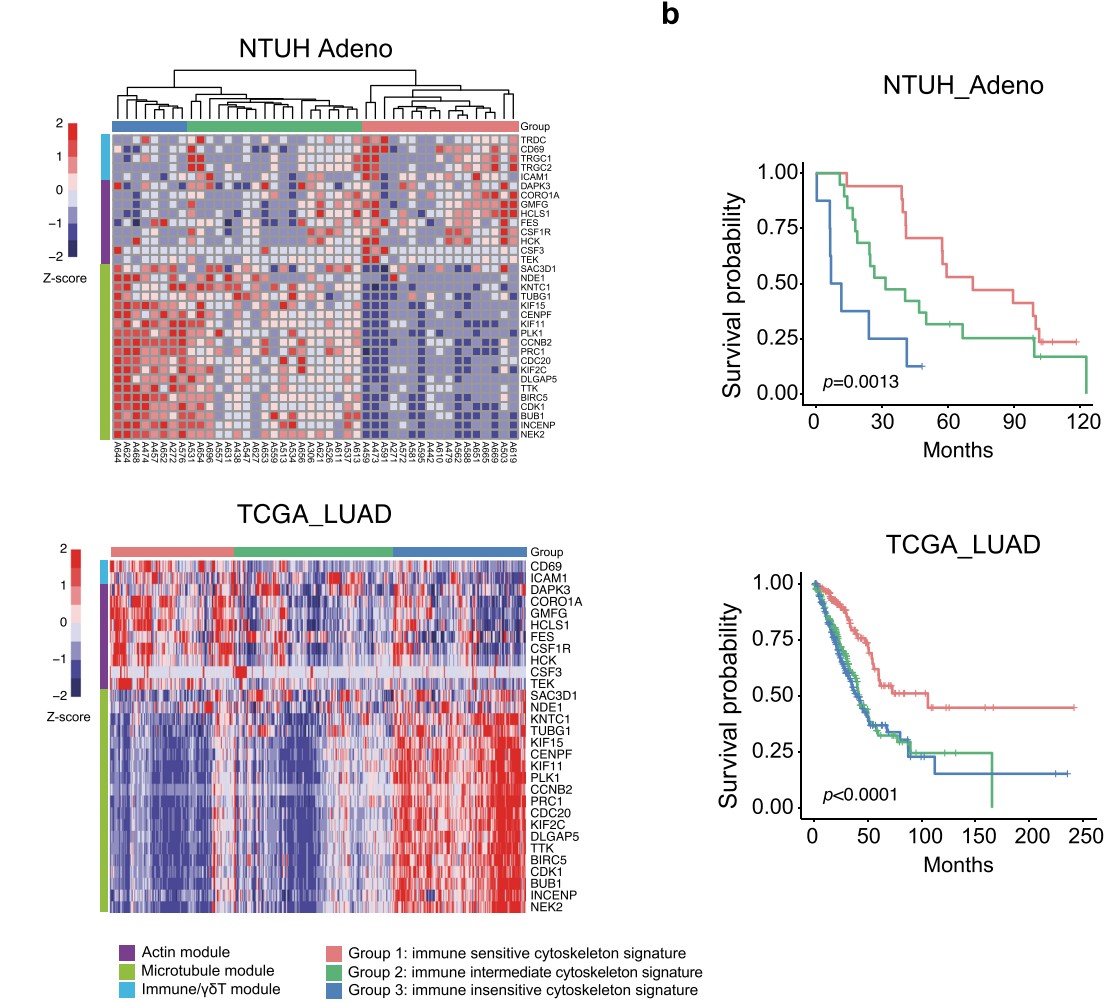

**Fig. 7 Stratification of patients with lung cancer by the immune cytoskeleton gene signature. a** Heatmaps of immune cytoskeleton gene signature derived from mRNA-seq data of primary lung adenocarcinoma tumor tissues in patients at National Taiwan University Hospital (NTUH, upper) and from The Cancer Genome Atlas (TCGA, lower). **b** Overall survival analysis of the NTUH (upper) and TCGA (lower) lung adenocarcinoma patient cohorts stratified by immune cytoskeleton gene signatures associated with different γδ T susceptibilities. The *p* value is calculated by the Mantel-Cox test (one-sided).

which do not rely on MHC-dependent cancer recognition, are on the rise[55,56]. Using a quantitative surface proteomics approach combined with epigenomic/transcriptomic analysis, we uncovered an immunomodulatory effect of DNMTis to facilitate tumor cytolysis by MHC-unrestricted γδ T cell-based therapy via upregulation of adhesion molecules and reorganization of the immunosynaptic cytoskeleton networks to strengthen the encounter between cancer and γδ T cells.

γδ T cells are a distinct subset of T cells that combine adaptive characteristics and rapid, innate-like responses. Two major subsets of human γδ T cells, Vδ1+ and Vδ2+, are defined based on the recombination of the TCR γ and δ chains. Vδ1+ T cells are predominantly located in peripheral tissues and possess robust anticancer capacities without the requirement for opsonizing antibodies, as Vδ2+ T cells do[57–59]. Nevertheless, the clinical use of Vδ1+ T cells has been limited by the lack of a reliable expansion protocol. Instead, Vδ2+ T cells, which can be easily expanded with amino bisphosphonates (e.g., zoledronic acid) or phosphoantigens, are used in most clinical trials with limited clinical efficacy against solid tumors[5,6]. Moreover, certain γδ T subsets may possess protumor functions[7]; therefore, caution should be taken during the process of cell expansion for therapeutic uses. In light of recent advances in the protocol for DOT cell

generation[26–28], our modified expansion protocol eliminates the need for αβ T depletion and enriches Vδ1+ T cells with antitumor immunity instead of the protumor IL-17-producing γδ T subsets that have been shown to promote tumor progression[60,61]. This clinical-grade expansion protocol for Vδ1+ cells may facilitate the refinement of current γδ T cell therapy, as well as the development of dual-functional chimeric antigen receptor-engineered human γδ T cells (CAR-γδ T cells) that respond to both stress signals and specific cancer antigens[62,63].

Notably, we show that DNMTis may remodel cancer cells' immune cytoskeleton or upregulate general adhesion molecules such as ICAM-1. These structural components have essential roles in normal physiology and are present in normal cells as well. Nevertheless, we did not observe significant toxicity or normal tissue damages in vitro or in vivo. It is likely that the potentiating effect of DNMTi is built upon the existing tumor-recognition capacities of γδ T cells. And the drug acts to strengthen the interaction between γδ T and cancer cells after the initial encounter takes place. In fact, we demonstrated that blocking of NKG2D attenuated γδ T-mediated cytolysis of DAC-pretreated H1299 lung cancer cells (Supplementary Fig. 8b), which suggests that conventional tumor recognition mechanisms by γδ T cells, such as NKG2D/NKG2DL interactions, partially contribute to γδ

T killing of DAC-treated cells[64]. On the other hand, epigenetic upregulation of NKG2DL on cancer cells may not be the sole determinant for the potentiation effects of DNMTi. As shown in our proteomics data across different lung cancer cell lines, the DNMTi-induced upregulation of γδ TCR or NKG2D ligands occur in some but not all cell lines in which the potentiating effect of DNMTis is observed. In addition, each γδ TCR subtype appears to have its own specific target ligands. It is less likely that one specific surface ligand accounts for the general potentiating effects of DNMTis on γδ T-mediated killing. Thus, our finding of DNMTi-mediated cytoskeletal reorganization via a coordinated epigenetic process provides a missing link to enhance the effectiveness of γδ T-based therapy.

Furthermore, our data of the immune synaptic-cytoskeletal networks on the target cell side of the immune synapse have highly significant clinical implications. Strikingly, primary human lung cancer tissues in two independent patient cohorts can be clearly stratified based on this epigenetic cytoskeletal gene signature. This implies that cytoskeletal arrangements are an intrinsic feature of cancer cells and may serve as a marker for the selection of patients who are likely to respond to cell-based immunotherapy directly or benefit from additional epigenetic remodeling. Thus, the possibility of epigenetically converting an immune-resistant cytoskeleton pattern into an immune-sensitive pattern may provide a potential therapeutic strategy to maximize the clinical benefits of cell-based immunotherapy.

Additionally, our holistic proteomics approach of characterizing the surface immunome of lung cancer cells reveals both innate and adaptive immune protein networks that can be altered by DAC and shed light on the drug's multifaceted immunomodulatory potentials to couple with various types of immunotherapy, including γδ T and NK cells, among others[64–67]. We also demonstrate that DAC may enhance the polyfunctionality of the Vδ1 subtype to exert antitumor immunity. This finding differs from a previous report showing that DAC may upregulate the inhibitory receptor KIR2DL2/3 on ex vivo expanded Vδ2 T cells and inhibit their proliferation and effector functions[68]. This disparity could be attributable to different γδ T cell subsets, expansion conditions, or DAC dosing schedules.

Likewise, the safety and efficacy of combined treatment of DNMTi and ex vivo expanded Vδ1 cells can be context-dependent. In particular, while we observed little toxicity on normal PBMCs (Supplementary Fig. 4e), the finding does not negate the potential off-target effects on other normal cell or tissue types, which should be taken into consideration when designing clinical trials. In addition, although we demonstrated the potentiating effect of DNMTi on γδ T-mediated tumor lysis in immunocompromised mouse model with little interference of other immune cell types, the model is limited in recapitulating the complex interactions between cancer cells and an intact immune microenvironment. Since ICAM-1 is a general adhesion molecule and may be recognized by other T cell subtypes, such as αβ T cells, it is possible that a similar ICAM-1-dependent mechanism may be involved in the interaction between cancer and αβ T cells in an MHC-restricted manner. This might potentially enhance the efficacy of immunotherapy by involving more T cells with cytolytic functions. The modest potentiation effect of DAC on CD8 T cells we observed (Supplementary Fig. 7) is probably due to alloreactive recognition. Further investigation on the roles of αβ T and other immune cell types in DNMTi-potentiated γδ T therapy is warranted.

In summary, our findings indicate the therapeutic potential of DAC in combination with adoptive immunotherapy with γδ T cells in lung cancer patients who are unresponsive to checkpoint inhibitors or lack targetable mutations/specific cancer antigens. In particular, those who have low ICAM-1-expressing tumors or unfavorable cytoskeleton gene signatures may benefit from epigenetic reprogramming to enhance the therapeutic efficacy of γδ T cells. This study also provides a molecular basis for pharmacologically modulating immune synaptic-cytoskeletal networks to achieve effective antitumor immunity, which may support delicate clinical trial designs and facilitate the development of therapeutic strategies for lung cancer in the era of precision immuno-oncology.

## Methods

**Cancer cell lines and drug treatment.** Human lung cancer cell lines, A549, H1299, HCC827, PC-9, PC-9-IR, H2981, H157, H1792, H2170, and colorectal cancer cell line, HCT116, were obtained from the American Type Culture Collection (ATCC). CL1-0 and CL1-5 human lung adenocarcinoma cell lines were kindly provided by Prof. Pan-Chyr Yang at the National Taiwan University College of Medicine[69] (Supplementary Table 1). Cells were grown in HyClone PRMI medium (SH30027.02, GE Healthcare) supplemented with 10% GIBCO dialyzed FBS (26140-079, Life Technologies), 1% L-glutamine (A29168-01, Life Technologies), and 1% penicillin-streptomycin (15140-122, Life Technologies). Authentication of all cell lines used in this study was performed using short tandem repeat (STR) analysis. For drug treatment experiments, cells were cultured with 100 nM decitabine (DAC; A3656, Sigma-Aldrich) for 3 days with daily change of complete medium and drug replenishment, followed by 3–4 days of cell recovery in the decitabine-free medium. To block the DAC's effect on immune synapse formation via actin cytoskeleton reorganization, the DAC-pretreated (D3R3) cells were treated with 1 μg/mL cytochalasin B (Cyto B; C6762, Sigma-Aldrich) for 1.5 h prior to coculture with γδ T cells.

**SILAC labeling.** For quantitative mass spectrometry analysis, lung cancer cells were cultured for at least seven cell doublings in SILAC RPMI medium (88365, Thermo Fisher Scientific) supplemented with 10% FBS and heavy isotope-labeled amino acids (L-Lysine 13C6, L-Arginine 13C6; CLM-2247-H-1, CLM-2265-H-1, Cambridge Isotope Laboratories). DAC-treated cells grown in the regular medium were considered as SILAC light counterparts.

**Biotinylation and isolation of cell surface proteins.** Cancer cells (~$1.5 \times 10^7$ cells) were grown in a 15-cm culture dish and washed with ice-cold PBS (21040CM, Corning) before the labeling. A total of 5 mg of membrane-impermeable sulfosuccinimidyl-6-(biotin-amido) hexanoate reagent at the concentration of 0.3 mg/mL (EZ-link sulfo-NHS-SS-biotin; 21331, Thermo Fisher Scientific) was applied to the cells with gentle shaking at 4 °C for 30 min. The reaction was quenched by 20 mM glycine (G8790, Sigma-Aldrich) in PBS for 10 min. Subsequently, biotinylated cells were collected by scraping in PBS containing 100 μM oxidized glutathione (G4376, Sigma-Aldrich) to prevent the reduction of the disulfide bridge in the labeling molecule. After centrifugation at $500 \times g$ for 3 min, the cell pellet was subjected to a freezing-thawing step and resuspended in the lysis buffer [2% Nonidet P40 (NP40; 11332473001, Roche), 0.2% SDS (75746, Sigma-Aldrich), 100 μM oxidized glutathione and 1X Halt protease inhibitor cocktail (87786, Thermo Fisher Scientific) in PBS)] for 30 min on ice. The lysate then underwent sonication with five 15-s bursts and was centrifuged at $14,000 \times g$ for 5 min at 4 °C to remove insoluble materials. The protein concentration of the supernatants was determined using Pierce 660 nm Protein Assay Reagent Kit (22660, Thermo Fisher Scientific) supplemented with Ionic Detergent Compatibility Reagent (22663, Thermo Fisher Scientific). The biotinylated protein extracts from unlabeled and SILAC-heavy-labeled cancer cells were mixed in 1:1 (w/w), and the mixture was subjected to biotin-affinity purification. Pierce NeutrAvidin-agarose slurry (29200, Thermo Fisher Scientific) in 200 μl/mg of total proteins was conditioned by three washes in buffer A (1% NP40 and 0.1% SDS in PBS). Binding of biotinylated proteins was performed on a rotating mixer at 4 °C overnight. The beads were then washed twice with buffer A, twice with buffer B [0.1% NP40, 10.3 M NaCl (4058-01, JT Baker) in PBS] and twice with buffer C (50 mM in PBS, pH 7.8). Proteins were eluted twice for 30 minutes each at 58 °C with 150 mM dithiothreitol (DTT; D0632, Sigma-Aldrich), 1% SDS, 50 mM Tris-HCl in PBS (pH 7.8). Subsequently, 150 mM iodoacetamide (IAA; I6125, Sigma-Aldrich) was added to the sample followed by incubation for 30 min at room temperature in the dark to alkylate reduced cysteine residues in proteins. To concentrate the proteins and reduce the detergents in the membrane-rich eluate, we purified the eluate using a Amicon MWCO-10K centrifugation unit (UFC501096, Millipore) with the exchange buffer [0.5% SDS in 50 mM NH$_4$HCO$_3$ (09830, Sigma-Aldrich) aqueous solution]. The sample retained in the exchange buffer was then recovered for subsequent analyses[70].

**In-gel digestion and mass spectrometry.** The purified biotinylated-surface proteins were fractionated using SDS-PAGE, followed by in-gel trypsin digestion[71]. The trypsinized peptides were vacuum-desiccated and solubilized in 0.1% trifluoroacetic acid (TFA; 299537, Sigma-Aldrich). The sample was then desalted using C18 Ziptip (ZTC18S960, Millipore) according to the manufacturer's

protocol. Peptide mass acquisition was performed on an Ultimate system 3000 nanoLC system connected to the Orbitrap Fusion Lumos mass spectrometer equipped with NanoSpray Flex ion source (Thermo Fisher Scientific). After loading the peptides into the HPLC instrument, the peptides were concentrated by a C18 Acclaim PepMap NanoLC reverse-phase trap column with a length of 25 cm and an internal diameter of 75 μm containing C18 particles sized at 2 μm with a 100 Å pore (164941, Thermo Fisher Scientific). The mobile phase aqueous solvent A (0.1% formic acid; 33015, Sigma-Aldrich) and organic solvent B [0.1% formic acid in acetonitrile (9829, JT Baker)] were mixed to generate a linear gradient of 2% to 40% solvent B for fractionated elution of peptides. The mass spectra were acquired from one full MS scan, followed by data-dependent MS/MS of the most intense ions in 3 s. Full MS scans were recorded with a resolution of 120,000 at $m/z$ 200. For MS/MS spectra, selected parent ions within a 1.4 Da isolation window were fragmented by high-energy collision activated dissociation (HCD) with charge status of 2+ to 7+. The duration of the dynamic exclusion of parent ions was set at 180 s with a repeat count. Mass spectra were recorded by the Xcalibur tool version 4.1 (Thermo Fisher Scientific).

**LC-MS/MS data analysis and SILAC-based protein quantitation**. The resulting Thermo RAW files were analyzed using MaxQuant v1.6.0.16[72]. MS/MS spectra were searched in the Andromeda peptide search engine against the UniProtKB/ Swiss-Prot human proteome database as well as the reversed counterpart as a decoy database. The common contaminants list provided by MaxQuant software was applied during the search. Peptides with a minimum of six amino acids and maximum trypsin missed cleavages of three were considered. The setting of variable modifications includes methionine oxidation, protein N-terminal acetylation, asparagine/glutamine deamidation, and EZ link on primary amines after sulfhydryl reduction and alkylation by IAA (EZ-IAA, +145.020 Da)[73]. Carbamidomethyl cysteine was applied as a fixed modification. Initial parent peptide and fragment mass tolerance were set to 20 ppm and 0.5 Da, respectively. False discovery rate (FDR) filtration of the peptide-spectrum match and protein assignment were utilized at 0.05 and 0.01, respectively. Finally, proteins identified as a reverse decoy, matched with only one unique peptide, and as common contaminations were excluded before further analysis.

**Patient-derived lung cancer cells from malignant pleural effusions**. The malignant pleural effusions from lung cancer patients were centrifuged at $1800 \times g$ for 5 min to collect cell pellets. Cell pellets were resuspended in 4 ml PBS and subject to density gradient centrifugation with Ficoll-Paque PLUS (17144002, GE Healthcare) according to the manufacturer's instructions. Briefly, the resuspended cells were carefully loaded onto 3 ml Ficoll-Paque PLUS in a 15-ml centrifuge tube and layered by centrifugation at $1800 \times g$ for 20 min at room temperature. Nucleated cells enriched at the interface were collected and washed by at least three volumes of PBS. Then the cells were pelleted by centrifugation at $300 \times g$ for 5 min. Red blood cells (RBC) in the cell pellet were lysed with RBC lysis buffer [155 mM $NH_4Cl$ (11209, Sigma-Aldrich), 10 mM $KHCO_3$ (2940-01, JT Baker) and 0.1 mM EDTA (34550, Honeywell Fluka) in deionized water], and discarded after centrifugation of the sample at $300 \times g$ for 3 min. The cell pellet was finally washed twice with PBS and centrifuged at $300 \times g$ for 3 min. The collected cells were grown in DMEM/F-12 (1:1 in v/v; 11330, Thermo Fisher Scientific) supplemented with 5% FBS, 2% penicillin-streptomycin, 0.4 μg/mL hydrocortisone (H088, Sigma-Aldrich), 5 μg/mL insulin (I2643, Sigma-Aldrich), 10 ng/mL epidermal growth factor (PHG0311L, Invitrogen), 24 μg/mL adenine (A2786, Sigma-Aldrich), and 6 μM Y-27632 (ALX-270-333, Enzo Life Sciences). Floating lymphocytes in the primary culture were eliminated by PBS washing before a detachment of adherent cells with trypsin-EDTA (25200072, Thermo Fisher Scientific) at 0.25% in PBS during each passage. The removal of fibroblasts was achieved due to their faster adhesion to the culture dish than tumor cells. We transferred the trypsinized-cell suspension to a new culture dish and let it sit at 37 °C until the two types of cells were separated from each other. After repeated subcultures, the purity of the tumor cell population was confirmed by measuring the surface expression of EpCAM by flow cytometric analysis[74].

**Isolation and ex vivo expansion of γδ T cells**. $1 \times 10^7$ Peripheral blood mononuclear cells (PBMC) obtained from healthy donors were seeded in each well coated with anti-TCR PAN γ/δ antibody (IMMU510 clone) using a six-well culture plate. The culture medium contained Optimizer CTS T-Cell Expansion SFM (A1048501, Thermo Fisher), 15 ng/mL IL-1β (AF-200-01B), 100 ng/mL IL-4 (AF-200-04), 7 ng/mL IL-21 (AF-200-21), and 70 ng/mL IFN-γ (AF-300-02, Peprotech). After seven days, the medium was changed to Optimizer CTS medium, 5% human platelet lysate (PLS1, Compass Biomedical), 70 ng/mL IL-15 (AF-200-15, Peprotech), and 30 ng/mL IFN-γ. These cells were harvested on Day 21 for subsequent experiments, including in vitro cytotoxicity and animal experiments. We have complied with all ethical regulations. Informed consent was obtained from individual healthy donors before enrollment. The study was approved by the Institutional Review Board (IRB) of National Taiwan University Hospital.

**Immunophenotypic and functional analysis of γδ T cells**. Ex vivo expanded γδ T cells were washed twice with PBS and stained with immunofluorescence

antibodies targeting the surface markers, including γδTCR Vδ1, γδTCR Vδ2, CD27, CD69, NKG2D, TGF-β1, and CD107a (Supplementary Table 2). Subsequently, γδ T cells were fixed and permeabilized using Cytofix/Cytoperm solution (554714, BD Biosciences) for 20 min at 4 °C for intracellular staining of cytokines, including IL-2, IL-10, IL-17A, IFN-γ, and TNF (Supplementary Table 2). Staining was performed at 4 °C for 30 min in the dark. The samples were washed and fixed with 100 μl of 1X IOTest3 Fixative Solution (A07800, BECKMAN COULTER) per well for at least 10 min at 4 °C. Cells were then resuspended in 300 μl PBS and analyzed using a BD LSR Fortessa flow cytometry (BD Biosciences). Acquired data were analyzed using FlowJo software (Tree Star). Gating strategies are either provided in-figure or in Supplementary Fig. 16. For polyfunctional response measurement, γδ T cells were stimulated with 30 ng/mL phorbol 12-myristate 13-acetate (PMA; P1585, Sigma-Aldrich) and 1 μg/mL ionomycin (I9657, Sigma-Aldrich) in the presence of monensin (00-4505-51, eBioscience) and brefeldin A (420601, BioLegend) for 4 h at 37 °C. After stimulation, γδ T cells were transferred to v-bottom 96-well plates and stained for surface markers and intracellular cytokines as described above. As an unactivated control, γδ T cells were incubated only with dimethyl sulfoxide (DMSO; D2650, Sigma-Aldrich), monensin, and brefeldin A before staining.

**Single-cell mass cytometry**. Samples were processed as described with few modifications[75]. Briefly, the cell samples were first stained for viability with cis-platin (201064, Fluidigm) and then fixed with 1.5% paraformaldehyde (15710, Electron Microscopy Sciences) at room temperature for 10 min followed by two washes with Cell staining medium (CSM) [PBS containing 0.5% bovine serum albumin (BSA; A3059, Sigma-Aldrich) and 0.02% sodium azide (S2002, Sigma-Aldrich)]. Formaldehyde-fixed cell samples were then subjected to pre-permeabilization palladium barcoding[76]. The barcoded samples were first incubated with anti-TCR Vδ 1-FITC for 30 min on ice, washed once with CSM, and then stained with metal-conjugated antibodies against surface markers for 1 h. After incubation, samples were washed once with CSM, permeabilized with 1x eBioscience Permeabilization Buffer (00-8333-56, Thermo Fisher Scientific) on ice for 10 min, and then incubated with metal-conjugated antibodies against intracellular molecules for 1 h. Cells were washed once with 1x eBioscience Permeabilization Buffer and then incubated at room temperature for 20 min with an iridium-containing DNA intercalator (201192 A, Fluidigm) in PBS containing 1.5% paraformaldehyde. After intercalation/fixation, the cell samples were washed once with CSM and twice with water before measurement on a mass cytometer (Fluidigm). Normalization for detector sensitivity was performed[77]. After measurement and normalization, the individual files were debarcoded[76] and gated according to Supplementary Fig. 2b. viSNE maps were generated using software tools available at https://www.cytobank.org/. For antibody conjugations, antibodies in carrier-free PBS were conjugated to metal-chelated polymers (Supplementary Table 3) according to the manufacturer's protocol. Antibody conjugation to bismuth was carried out as previously described[78].

**Data analysis for mass cytometry data**. For the comparison between ex vivo expanded γδ T cells with and without DAC treatment, 50,000 cells from each group were randomly sampled and pooled together for clustering using X-shift[79], a density-based clustering method. All markers except the ones used for gating were selected for clustering. Clusters separated by a Mahalanobis distance <2.0 were merged. The optimal nearest-neighbor parameter, $K$, was determined as 20 using the elbow method. The expression level and the cell frequency in each cluster were exported and represented by heatmaps and piecharts using R. For the correlation between expression levels of markers in CD3+ T cells at baseline and after ex vivo expansion, pairwise Pearson correlation coefficients were calculated. The heatmap was generated and clustered based on hierarchical clustering of the Pearson correlation coefficients by R.

**γδ T-cell-mediated cytotoxicity assays**. Cancer cells were cocultured with γδ T cells at an effector to target (E:T) ratio of 3:1 at 37 °C for 2 h. After coculture, cell death was evaluated by flow cytometric analysis using FITC Annexin V Apoptosis Detection Kit I (556547, BD Biosciences). In addition, we performed real-time monitoring of γδ T-mediated killing of cancer cells using the Electric Cell-Substrate Impedance Sensing (ECIS) monitoring system with an 8W10E + culture chamber (Applied Biophysics). Mock or DAC-treated cancer cells were cultured in the chamber at 37 °C overnight until the cancer cells were fully attached to the bottom of the wells. Following the addition of γδ T cells, the detachment of cancer cells indicating cell death was recorded in real-time using multiple frequency capture with the ECIS software. Relative impendence at the time of γδ T cell addition was used for between-sample normalization. For non-contact killing experiments, cancer cells were seeded in the bottom wells of a 24-well Transwell system overnight, followed by the addition of γδ T cells onto the top 0.4 μm pore membrane inserts that are impermeable to cells (353095, Falcon). The coculture was performed at an E:T ratio of 10:1 at 37 °C for overnight. The death of cancer cells in the bottom wells was evaluated by flow cytometric analysis using FITC Annexin V Apoptosis Detection Kit I.

**γδ T cell chemotaxis assay**. Cancer cells were cocultured with γδ T cells in a Transwell system with a 3 μm pore membrane insert (3415, Falcon) that allows γδ T cells to pass through. Prior to the coculture, cancer and γδ T cells were stained with vital dyes, Calcein AM (1755, BioVision), and Hoechst 33342 (H3570, Thermo Fisher Scientific), respectively. After coculture for 2 h, the γδ T cells (positive for Hoechst 33342) that had migrated into the bottom chamber were imaged under a fluorescence microscope and quantified by ImageJ.

**Immunofluorescence imaging of immune synapses**. Cancer and γδ T cells were spun down onto poly-L-lysine coated coverslips by using a Cyto-Tek table-top cytofuge at 500 rpm for 5 minutes (Sakura Scientific) before fixation with ice-cold methanol or 4% (w/v) paraformaldehyde in PBS for 10 min. The fixed samples were blocked in detergent-free blocking solution [1% normal donkey serum (ab7475, abcam) and 3% BSA in PBS] without membrane permeabilization by detergents. Subsequently, cells were incubated with primary antibodies diluted in the blocking solution for 1 h in a moist chamber at room temperature and washed with PBS. Labeling of fluorescent secondary antibodies at 1:200 dilution in PBS was carried out in the blocking buffer for 1 h at room temperature. After PBS washing, cell nuclei were stained with Hoechst 33342 or DAPI (62248, Thermo Fisher Scientific) in PBS. Finally, the samples were washed with PBS and mounted on the slides. All slides were examined under an epifluorescence microscope EVOS FLc (Invitrogen), and the images were analyzed using the ImageJ software. For the visualization of immune synaptic proteins, cells were imaged with a high-resolution confocal microscope (LSM780, Zeiss) with a ×63 oil objective and analyzed with the confocal software ZEN (Zeiss). Antibodies used in the study are listed in Supplementary Table 2.

**Overexpression and CRISPR/Cas9 gene knockout experiments**. In overexpression experiments, doxycycline-inducible ICAM1 overexpressing vector was created through the cloning of full-length ICAM1 cDNA into a Tet-On lentiviral plasmid pLVX-Tight-Puro (632162, Clontech). The gene expressing vector and the regulator vector (pLVX-Tet-On Advanced) were packaged with VSV-G pseudo-typed lentivirus particles in 293 T cells. After co-transfection of the two lentiviral particles into lung cancer cells, the cells were then grown in the selection media containing G418 (10131035, Thermo Fisher Scientific) and puromycin (A1113803, Thermo Fisher Scientific) at proper concentrations for the retention of both plasmids. In knockout experiments, editing of the ICAM1 genome locus on H1299 cells was achieved through coexpression of the Cas9 protein with the guide RNAs (gRNAs) targeting to ICAM1 exon 2 at the sequences 5′- TCAAAAGTCATCC TGCCCCG -3′ and 5′- GTGACCAGCCCAAGTTGTTG -3′. ICAM1-null cell lines were established through clonal propagation from single cells. The overexpression and loss of ICAM-1 protein were validated by flow cytometry with an anti-ICAM1 antibody (BBA20, R&D Systems). Besides, the edited genome patterns at ICAM1 locus around the gRNA-targeting sites of the ICAM1-knockout lines were validated by Sanger sequencing after PCR amplification (see Supplementary Table 4 for the PCR primer pairs). Analysis of the sequencing results for the CRISPR/Cas9-edited ICAM1 genome locus is performed using multiple sequence alignment tool (ClustalO) on QIAGEN CLC Genomics Workbench software (v20.0.2).

**RNA preparation and mRNA-seq analysis**. Total RNA was extracted with the PureLink RNA Mini Kit according to the manufacturer's instructions (12183018A, Invitrogen). The quality of RNA was evaluated using a Bioanalyzer 2100 with RNA 6000 Nano LabChip kit (5067-1511, Agilent Technologies). mRNA-seq libraries were prepared using TruSeq Stranded mRNA Library Prep Kit (RS-122-2101, Illumina) and sequenced using the HiSeq 4000 system. Raw reads were processed with adapter trimming and quality filtering using Trimmomatic with default settings[80]. The cleaned reads were aligned to UCSC human genome hg19 using RSEM tool with the bowtie2 aligner[81,82]. Mapped reads were counted for each gene using the R packages GenomicFeatures (v1.36.4) and GenomicAlignments (v1.20.1), according to the GENCODE human GRCh37 annotation (https://www.gencodegenes.org/human/release_25lift37.html)[83]. Finally, FPKM normalization of the raw-counts was performed using DESeq2 (v1.24.0)[84]. Bar graphs and dot plots of RNA-seq data were created by ggplot2 (v3.2.1). Networks and upstream regulator analysis were conducted by Ingenuity Pathway Analysis (IPA; version 01-16, Qiagen).

**Gene ontology and gene set enrichment analysis**. Surface proteins upregulated by more than 1.4-fold with DAC treatment in the quantitative membrane proteomic analysis were subjected to gene ontology (GO) analysis using the AmiGO 2 web tool, PANTHER (v2.5.12), with Fisher's Exact test. For gene set enrichment analysis (GSEA), we used mRNA-seq data of cancer cell lines with and without DAC treatment to identify gene sets enriched in the DAC-treated samples (FDR < 0.25 and nominal p-value < 0.05). Cytoskeleton-associated gene sets are retrieved from the Molecular Signatures Database (MSigDB, v6.2).

**Genome-wide DNA methylation analysis**. Genomic DNA of cancer cells was extracted with the QIAamp DNA Mini Kit (51304, QIAGEN) according to the manufacturer's instruction. The DNA concentration and quality were evaluated by NanoDrop 2000 (Thermo) and electrophoresis with 0.8% agarose gel, respectively.

Bisulfite conversion of 1 μg genomic DNA was performed using EZ DNA Methylation Kit (D5001, Zymo Research). The bisulfite-converted DNA samples were subjected to genome-wide methylation analysis using the Illumina Infinium MethylationEPIC BeadChips. Raw intensity data were obtained as IDAT files and processed using R package minfi v1.30.0[85] with a probe annotation package for Illumina EPIC array (IlluminaHumanMethylationEPICanno.ilm10b4.hg19). The data were quantile normalized using the preprocessQuantile function of minfi. We removed low-quality probes with detection P-value > 0.01 as well as probes described as single nucleotide polymorphisms (SNPs), cross-reactive and genetic variants[86,87]. Finally, a total of 692,476 probes were used for further analysis.

**Genome-wide chromatin accessibility analysis**. Chromatin accessibility of lung cancer cells before and after DAC treatment was analyzed using the Omni-ATAC protocol[43] with modifications. After harvesting cells with trypsin/EDTA, a total of $1 \times 10^5$ cells were resuspended in 1 ml of cold ATAC resuspension buffer [ATAC-RSB; 10 mM Tris-HCl pH 7.4, 10 mM NaCl, and 3 mM $MgCl_2$ (AM9530G, Thermo Fisher Scientific) in water] supplemented with 0.1% Tween-20 (P2287, Sigma-Aldrich). Cells were centrifuged at $650 \times g$ for 5 min at 4 ℃ to remove the buffer and lysed in 50 μl of ATAC-RSB containing 0.1% NP40, 0.1% Tween-20, and 0.01% digitonin (G9441, Promega) on ice for 3 min. Subsequently, the lysate was washed by 1 ml of ATAC-RSB containing 0.1% Tween-20, and the supernatant was discarded following centrifugation at $650 \times g$ for 10 min at 4 ℃ to pellet the cell nuclei. The nuclear fraction was then resuspended in 50 μl of 1:1 (v/v) premix of 2X Tagmentation DNA (TD) buffer [20 mM Tris-HCl, 10 mM $MgCl_2$, 10% dimethyl formamide (D4551, Sigma-Aldrich), pH 7.6] and Transposition buffer [2 μl of Illumina adapters-bearing Tn5 transposase (FC-121-1030, Illumina), 0.02% digitonin and 0.2% Tween-20 in PBS]. The sample was incubated at 37 ℃ for 30 min in a water bath and vortexed every 5 min to facilitate the transposition reaction. At the end of the reaction, the transposed DNA fragments were harvested and cleaned up with Zymo DNA Clean and Concentrator-5 kit (D4011, Zymo Research). The libraries were indexed and amplified with Illumina i5/i7 indexing primers (FC-121-1011, Illumina) by PCR reaction to reach a target concentration of 4 nM in 20 μl. Library quality was checked by Agilent 2100 Bioanalyzer with high sensitivity DNA kit (5067-4626, Agilent Technologies). The sample was sequenced on the Illumina HiSeq X Ten platform with 151 bp paired-end sequencing for an average of 60 million raw reads per sample.

**Bioinformatic analysis of the Omni-ATAC-seq data**. The Nextera adapter sequence (5′-CTGTCTCTTATACACATCT-3′) was trimmed from the raw reads using CutAdapt v2.7[88]. Trimmed reads of each sample were mapped to the human reference genome build UCSC hg19 using BWA mem v0.7.17-r1188[89] with default -M parameter and a maximum fragment length of 2,000. The PCR duplicated reads were marked using Picard (v2.21.4, http://broadinstitute.github.io/picard/). Further quality filtering was performed using SAMtools (v1.9, PMC2723002) to remove unmapped reads, unmapped mates, PCR duplicated reads, unpaired alignments, and reads mapped to mitochondrial DNA. The size distribution of the filtered reads and nucleosome-occupancy frequency was evaluated by ATACseqQC v1.8.5[90]. The broad peak regions of ATAC-seq were called using MACS2 v2.2.5[91] with parameters: --nomodel --shift -75 --extsize 150 --keep-dup all --broad --broad-cutoff 0.1. Annotation of the peaks and the distance of each peak to the closest TSS were determined using ChIPseeker v1.20.0[92]. The bam files of the post-filtering read alignment from mock- and DAC-treated samples were simultaneously normalized with reads per genomic content (RPGC) approach by --effectiveGenomeSize 2827437033 and the log2 ratio of DAC/mock per bin was reported using deepTools v3.3.1[93].

**Transcriptomic data of primary lung cancer tissues**. The genome-wide gene expression data by mRNA-seq were obtained from two lung adenocarcinoma cohorts, National Taiwan University Hospital (NTUH), and the Cancer Genome Atlas (TCGA) database. The NTUH data were generated by our laboratory and deposited in the Gene Expression Omnibus (GEO) database with an accession number of GSE120622. The curated TCGA lung adenocarcinoma (LUAD) data of normalized gene expression (RNASeq2GeneNorm) and clinical information were acquired using an R package, curatedTCGAData v1.6.0[94]. Heatmaps of the Z-transformed gene expression level of selected genes of NTUH and TCGA mRNA-seq data were created using the R package pheatmap (v1.0.12).

**Combination therapy of DAC and γδ T cells in a xenograft mouse model**. Six-week-old male NOD.Cg-PrkdcscidIl2rgtm1Wjl/SzJ (NSG) mice were purchased from the National Laboratory Animal Center (Taiwan) and maintained under the standard pathogen-free condition. All mice were housed in an AAALAC accredited animal facility with a 12-h dark/light cycle (8 a.m. to 8 p.m.) at a temperature between 20 and 24 ℃ and a humidity between 50 and 70%. H1299 lung cancer cells were injected subcutaneously into the mice ($1 \times 10^7$/mouse). The therapy was started at seven days post-injection. For each two-week treatment cycle, tumor-bearing mice were treated with DAC (0.2 mg/kg BW) by intraperitoneal injection on Days 1, 2, and 3. Human γδ T cells ($1 \times 10^7$/mouse) were intravenously injected via tail vein on Days 5 and 10. Mice were monitored once per week and tumor sizes were measured by a digital caliper. Experiments were terminated when tumor sizes

reached 20 mm or weight loss by 20% or more was observed according to the institutional ethical board. The survival time of individual mice in each treatment group was recorded. The surviving mice on day 42 post-tumor injection were killed to obtain tumors for pathologic examination and hematoxylin and eosin (H&E) staining. For in vivo imaging of γδ T cell trafficking, γδ T cells ($1 \times 10^6$ cells) were prestained by CYTO-ID Red long-term cell tracer dye (Enzo Life Sciences, Farmingdale, NY). H1299 lung cancer cells were subcutaneously injected into the NSG mice. One week later, the tumor-bearing mice were treated with DAC (0.2 mg/kg BW) by intraperitoneal injection for 3 consecutive days (days 1–3) followed by tail vein injection of the prestained γδ T cells ($1 \times 10^7$/mouse) on day 5 and day 12. The images were taken at 2 and 4 h after γδ T injection using the IVIS Spectrum (Ex570, Em640) on day 12. All mice experiments were approved by the NTU College of Medicine Institutional Animal Care and Use Committee (IACUC; Protocol #20180077) and we have complied with all ethical regulations for animal testing and research.

**Statistical analysis**. Statistical analysis was performed using GraphPad Prism 8 and computing environment R. Two-sided Mann–Whitney or unpaired $t$-tests were used to compare the means between two groups, whereas one-way ANOVA with Tukey's multiple comparisons was used to compare the means of three or more groups. Overall survival analysis for the NTUH and TCGA lung adenocarcinoma cohorts was performed using the Cox regression model, as well as the Kaplan–Meier method. Calculation and plotting of the survival curve were performed using R packages survival (v3.1-8) and survminer (v0.4.6), respectively. Heatmaps of the Z-transformed gene expression level of mRNA-seq data were created by using R package pheatmap (v1.0.12). Bar and spot charts representing methylation ß-value and RNA-seq FPKM were created by ggplot2 (v3.2.1). All software programs used in this study are listed in Supplementary Table 5.

**Reporting summary**. Further information on research design is available in the Nature Research Reporting Summary linked to this article.

## Data availability

Genomic data generated in this study are available in the Gene Expression Omnibus (GEO) database under the accession numbers: "GSE145588 (genome-wide methylation)", "GSE145663 (mRNA-seq for lung cancer cell lines)", and "GSE145663 (Omni-ATAC-seq)". The mRNA-seq data of colorectal cancer cell lines with DNMT depletion can be accessed at "GSE93136". The mRNA-seq data of human lung cancer tissues in the NTUH cohort can be accessed at "GSE120622". The CyTOF raw FCS files have been deposited to FlowRepository database with the identifier "FR-FCM-Z2G5". The proteomics data have been deposited to the ProteomeXchange Consortium with the dataset identifier "MSV000084997" through the MassIVE partner repository. Human reference proteome can be accessed at "UniprotKB" [ftp://ftp.uniprot.org/pub/databases/uniprot/current_release/knowledgebase/reference_proteomes/Eukaryota/]. Full microscopy image data sets have been deposited to Mendeley Data (DOI: 10.17632/cx2mxszth9.1). Source data are provided with this paper.

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

## Acknowledgements

We sincerely thank Dr. Stephen Baylin at Johns Hopkins University School of Medicine for his feedback on the manuscript. We acknowledge the service provided by the Flow Cytometric Analyzing and Sorting Core and the Imaging Core of the First Core Laboratory and the Laboratory Animal Center at National Taiwan University College of Medicine. We would also like to thank the Tai-Cheng Cell Therapy Center and the Instrumentation Center of National Taiwan University for technical assistance. Mass cytometry analyses were performed by GRC Mass Core Facility of Genomics Research Center, Academia Sinica, Taipei, Taiwan, and Stanford Human Immune Monitoring Center, USA. This study was supported by Ministry of Science and Technology (MOST 105-2628-B-002 -040, MOST 108-2314-B-002-094), National Health Research Institutes

(NHRI-EX106-10610BC), Excellent Translational Medical Research Grant by National Taiwan University College of Medicine and National Taiwan University Hospital (105C101-71, 106C101-B1), National Taiwan University and Academia Sinica Innovative Joint Program (108L104303, 109L104303), as well as Institutional Top-down Research Grant by National Taiwan University Hospital (NTUH 107-T10, 108-T10, 109-T10).

## Author contributions

Conceptualization by H-C.T., R.R.W., and T-C.H.; methodology by R.R.W., C-T.L., H-C.T., and S-Y.C.; software and formal analysis by R.R.W., H-C.T., S-Y.C., and S-Y.L.; investigation by R.R.W., H-H.L., C-C.F., R-S.L., Y-J.H., Y-H.J., Y-C.W., Z-C.H., X-H.L., T-Y.C., W-C.H., and C.L.; resources by C-T.L., C-J.Y., J-C.L., Y-L.C., and H-C.T.; writing – original draft by R.R.W., S-Y.C., and H-C.T.; writing – review & editing by H-C.T.; supervision by C-J.Y.; project administration by Y-J.H.; funding acquisition by H-C.T.

## Competing interests

The authors declare no competing interests.
