## [Peer Review File · Nature Communications]

REVIEWER COMMENTS

Reviewer #1 (gamma delta T cells, tumours) (Remarks to the Author):

In this study, Weng and colleagues propose a new role for DNA methyl transferase inhibitors (DNMTis) in cancer immunotherapy. They have specifically addressed if DNMTis such as decitabine (DAC) can reshape surface proteins of cancer cells to facilitate MHC-unrestricted recognition by gamma-delta (gd) T cells. They have chosen a cutting-edge protocol that enables the selective enrichment and enhanced antitumor functions of the Vd1 subset of human gd T cells, Delta One T (DOT) cells. They show that DOT cells are significantly more efficient against lung cancer cells when these are pre-treated with DAC, which orchestrates cytoskeleton rearrangements that stabilize and optimize (in terms of key surface protein expression, both for NK receptors and cellular adhesion, like ICAM-1) the immune synapse between DOT cells and cancer cells. Furthermore, the characterized DAC-associated cytoskeleton signature identifies a subset of lung cancer patients with improved survival, and may be useful for patient selection in future clinical trials. These results are novel and of substantial interest to the community.

The study uses cutting-edge molecular methodologies like SILAC and NGS (mRNA-seq and ATAC-seq); robust cell biology (including microscopy-based) approaches; and well established functional assays for T cells, in vitro and in vivo, that provide solid pre-clinical proof-of-principal for their proposition and its implications for cancer immunotherapy. The comprehensive molecular and cell biology assays performed to understand DAC modulation of the actin cytoskeleton are particularly impressive and worthy of publication in a top-tier journal.

The manuscript is well written and easy to follow. Results are in general discussed properly and the references are appropriate, with one outstanding exception mentioned below.

Major issue:

(Pages 7-8) The authors exaggerate the "improvement" of their expansion/ differentiation protocol compared to the published "DOT-cell protocol", while ignoring the most recent publication by that group (Di Lorenzo et al. Cancer Immunol Res 2019) where they further optimized "DOT cell production". So the authors need to clearly state what they have changed in comparison to the 2019 version of the DOT-cell protocol, namely by not depleting ab T cells. Critically, the expansion, enrichment and phenotype described here is completely in line with the characterization of DOT cells. Therefore, for the benefit of the readers, I suggest using the terminology DOT cells, instead of gd T cells, straight after introducing the expansion protocol, since there are various therapeutic gd T cell products being explored for cancer immunotherapy (see ref. 3, Sebestyen et al. 2020), and it is important to clarify that the authors are employing DOT cells, especially as these are planned to be tested in upcoming clinical trials.

Minor issues:

1. Summary: when referring to a "clinical-grade protocol to enrich for the Vd1 subset", the authors should add its known designation and also mention its enhancement of effector functions, thus changing to: "clinical-grade protocol to enrich for the Vd1 subset (Delta One T cells) while enhancing their antitumor effector functions". This should also feature in the title on page 7, which should be modified to: "Ex vivo expansion of Vg1-enriched gd T cells with enhanced antitumor effector functions".
2. The way the experiment in Fig 1 is set up is not very clear, from the Mock:DAC 1:1 mixture (Fig 1a) used for SILAC, to the analysis (Fig. 1b-h) focused on comparisons at D3 and D3R3 (after 3 days after drug removal). Can the authors clarify this better in the text?
3. When mentioning NKG2D ligands and their roles on gd T cell activation (page 6), a more specific review on the topic should be cited, such as Silva-Santos, Strid, *Frontiers Immunol* 2018; or Simões, Di Lorenzo, *Frontiers Immunol* 2018.
4. (Page 11) When stating "putative ligands for $\gamma\delta$ TCR, including CD1d complexes³⁰, BTN3A31, BTNL332 and others³³, are not consistently upregulated by DAC", the authors should qualify that such ligands, maybe with the exception of CD1d for a very minor subset, are NOT applicable to

Vd1 T cells.

5. Figure 6 is of limited interest (only to highly specialized readers) and should be moved to supplementary material.

6. By contrast, Figure 7 is very relevant but has 2 very distinct messages and sets of data, that should be split into 2 main figures (even if each of them small, which should not be an issue): one with the mouse xenograft model and another with the lung cancer patient stratification.

7. The authors do not show DOT-cell infiltration (into tumor lesions) nor persistence in the blood in their xenograft model (Fig 7a-c). Could they provide (or comment on) such data?

8. Besides the cytoskeleton gene signature, can the authors detect any DOT-cell associated markers in the primary lung cancer tissue analysed in Fig. 7d?

9. The discussion has some inaccuracies that should be corrected (pages 17-18):

➤ Remove intraepithelial from Vd1+ T cells, since they also circulate (that's why the authors could expand them from blood!); the following sentence already highlights their preferential tissue localization;

➤ Modify "the clinical use of Vδ1+ T cells has been fairly limited" to "the clinical use of Vδ1+ T cells has been limited by the lack of a reliable expansion protocol" and mention here the recent advances provided by the DOT-cell protocol (up to its 2019 version, see above), clearly stating that it was the basis for the current study (with minor modifications as clarified above, see major issue).

Reviewer #2 (IEL, cancer) (Remarks to the Author):

The present manuscript by Rueyhung R. Weng et al., entitled Epigenetic modulation of immune synaptic-cytoskeletal networks potentiates $\gamma\delta$ T cell-mediated cytotoxicity in lung cancer, provides new important information about therapeutic potential of human $\gamma\delta$ T cells against tumor cells treated with epigenetic modulation of DNA methyltransferase inhibitors. It is a well-designed and performed study and, overall, the manuscript is well written and organized in a systematic manner with solid and convincing experimental results. However, quite a few issues need further clarification and/or experimental data. The major concern stands from the authors statement that "ex vivo expanded human Vδ1-enriched $\gamma\delta$ T cells retain antitumor effector functions". This indeed suggest "a Vd1 T cell-dependent anti-tumor response against DEC-treated cancer cells". Authors should better state and rephrase

Major point:

1. As shown in Fig. 2a and Supplementary Fig 2c, enrichment of $\gamma\delta$ T cells accounts about 80 % of total CD3 T cells, in addition Vd1 T cells accounts approximately 70% of all $\gamma\delta$ T cells. Thus, it is not clear whether apoptotic and cytotoxic effects shown in Fig. 2 are due to the specific Vd1 or either Vd2 T cell subset. Moreover, other cells such as CD8 T cells could contribute to the killing of tumor cells. Thus, to directly assess $\gamma\delta$ T cell-mediated and in particular the specific Vd1-dependent cytotoxic response the authors should set up experiments that would discriminate cytotoxic response and cytokine (TNF- α , IFN- γ) production of different Vd1, Vd2 and CD8 T cell populations (e.g. CD107a flow cytometry-based assay).

2. Similarly, in Fig. 4 showing immunofluorescence staining of synapse formation the author should rather use FACS-sorted $\gamma\delta$ T or Vd1 T cells and not Vd1-enriched cells.

Minor points:

1. No statistically significant expansion of total $\gamma\delta$ T cells and of the specific Vd1 T cell-enrichment is shown in Fig. 2a.

2. Some figures (e.g. Fig. 2a or Fig. 5e) lack numbers of biological/experimental replicates.

Reviewer #3 (gamma delta T cells) (Remarks to the Author):

Weng et al present a study that focusses on pharmacological manipulation of $\gamma\delta$ T cell reactivity. This is of significant interest, not least because of the increasing focus on developing $\gamma\delta$ T cell-based immunotherapies, but has largely not been addressed previously outside of drugs (such as aminobisphosphonates) that influence phosphoantigen-mediated activation of V γ 9V δ 2 T cells. The study instead focusses on DNMTi drugs, some of which have been approved for use in certain haematological malignancies (AML, MDS). While their clinical impact is variable between patients and their mode of action is unclear, they have been suggested to exert a range of immunomodulatory effects.

Focussing largely on lung cancer, Weng and colleagues extend this body of knowledge by providing evidence that DNMTi treatment upregulates molecules involved in $\gamma\delta$ T cell activation on cancer cells, increases synapse formation between $\gamma\delta$ T cells and cancer cell targets, and potentiates $\gamma\delta$ T cell anti-tumour immunity, specifically involving V δ 1 rather than V γ 9V δ 2 T cells. Finally, they also suggest that lung cancer patients can be stratified by immune cytoskeleton expression signatures that relate to $\gamma\delta$ T cell-mediated cytotoxicity.

Strong points

The study has several strong points. The identification of drugs that boost V δ 1 tumour recognition (extending pharmacological manipulability beyond the well-established aminobisphosphonate/V γ 9V δ 2 axis), and that can be combined with V δ 1 T cell adoptive transfer, is as far as I am aware novel. In doing so the study brings to light a novel immunomodulatory aspect of DNMTi drugs, although the authors note that others have previously reported DNMTi enhancement of V γ 9V δ 2 cytotoxic responses to osteosarcoma cells (Wang et al, 2018). The current study also nicely applies a range of complementary techniques (SILAC, mass cytometry, microscopy, mouse models) to shed light on some mechanistic aspects of this enhanced V δ 1 tumour targeting. Of note, the importance of ICAM-1 in $\gamma\delta$ T cell recognition of target cells is not novel, and has been previously highlighted, both in the context of V δ 2-negative $\gamma\delta$ T cell recognition of infected and cancerous cells (eg Willcox, C. R. et al. Cytomegalovirus and tumor stress surveillance by binding of a human $\gamma\delta$ T cell antigen receptor to endothelial protein C receptor. *Nat. Immunol.* 13, 872–879 (2012).) and also discussed in reviews (eg Willcox CR and Willcox BE, *Nat Immunol*, 2019). However, the finding that this $\gamma\delta$ T cell/ICAM-1 axis is pharmacologically manipulable is novel. In light of the growing interest in $\gamma\delta$ T cell immunotherapies against cancer, my feeling is the study's findings will be of substantial interest to the field.

Criticisms

1. In some effector assays (eg Figure 2e, f) the impact of combining DAC-treatment of cancer cells and $\gamma\delta$ T cells is relatively weak, and appears to be largely additive (similar to or slightly more than the sum of the separate effects of DAC and $\gamma\delta$ T cells); moreover in at least one case the effect of combination even appears sub-additive (eg Figure 2f). However if DAC treatment was truly substantially boosting $\gamma\delta$ T cell recognition surely one would expect a synergistic effect? In contrast, in the animal model (eg Figure 7a,b) the effect of combining both DAC and $\gamma\delta$ T cells does appear to be synergistic. This 'additive effect' vs 'synergism' issue should be directly discussed – including in relation to these different experiments, and with reference to likely underlying mechanisms.

2. Regarding the V δ 1-focussed $\gamma\delta$ expansion protocol, I have two queries.

a) Aside from the obvious strong similarities to the expansion procedures developed by Silva-Santos and colleagues (which could be acknowledged more clearly), a second question is the extent to which the procedures as outlined preferentially enrich for V δ 1 T cells. In Figure 2a (and Supplementary Figure 2a), when the V δ 1-focussed expansion protocol is highlighted, the $\gamma\delta$ T cell compartment of the individual(s) at day 0 appears to atypical in that it is dominated already by V δ 1 T cells. This is quite unusual, as in most individuals V γ 9V δ 2 T cells predominate, and also the

resulting V δ 1-dominated post-expansion proportions of $\gamma\delta$ subtypes only reflect a slight enrichment of V δ 1 T cells compared to the starting profile. It would be more informative to show expansion from an individual with a more typical V δ 2-dominated $\gamma\delta$ T cell compartment, particularly as the text highlights the procedure's potential to elicit preferential V δ 1 enrichment. Based on 'normal' relative proportions of V δ 2 vs V δ 1 T cells at day 0 (usually around 4 or 5 to 1), if the same modest levels of V δ 1 enrichment were observed, it is quite likely that the $\gamma\delta$ T cell profile of most individuals following the three-week expansion procedure would actually be dominated numerically by V δ 2 T cells. The authors should therefore show expansions in a range of donors, including those with more normal ratios of V δ 2 to V δ 1 T cells.

b) Particularly in light of the point above (in a), which suggests that mixed populations of different $\gamma\delta$ T cell subsets have been used, the results are weakened significantly by not resolving in many assays which $\gamma\delta$ T cell subsets the readouts relate to. This is relevant to in vitro cytotoxicity results in Figures 2 and 3, synapse formation in Figures 4 and 5, and even the in vivo data in Figure 7. Given potentially overlapping effector functions of different $\gamma\delta$ T cell subsets (eg cytotoxicity and cytokine production), and the desire to obtain mechanistic clarity, this is suboptimal, and means it is not possible to attribute drug-induced effects to a specific $\gamma\delta$ T cell subset or collectively to the entire $\gamma\delta$ T cell compartment. The authors should include at least some assays where they can unequivocally resolve which subset the assay readouts relate to.

3. The increase in tumour-targeted $\gamma\delta$ T cell reactivity appears to be entirely dependent upon and largely involving ICAM-1 upregulation. Three key considerations follow from this.

a) The specificity of the approach for targeting of tumour cells versus normal cells is unclear. Potentially, ICAM-1 levels could comprise one component of a stress signature sensed by V δ 2-negative T cells. However, one concern would be that use of DNMTi drugs such as DAC may induce $\gamma\delta$ reactivity to non-cancer cells by upregulating ICAM-1 and strengthening synapse formation. This is not addressed as almost all analyses relate to tumour cell recognition, however could ultimately result in off-tumour toxicity. Consistent with this, clinical use of DNMTi drugs (eg in AML) can result in various side effects – it is unclear if these are immune-mediated. The authors comment in the Discussion that no toxicity was observed in the mouse model, but this is a xenograft model where many potentiating receptor/ligand interactions between V δ 1 T cells and mouse target cells will be mismatched. In my opinion the study would therefore be strengthened by any in vitro experiments on non-transformed human cells that address this angle.

b) Related to (a), it is unclear whether ICAM-1 induction on tumour cells over-rides signals other immune receptors often thought critical to $\gamma\delta$ T cell recognition, such as the $\gamma\delta$ TCR or NKRs, or whether such pathways are still critical but crucially potentiated by increased ICAM-1 on target cells. The authors hint at this in the Discussion when they highlight 'the potentiating effect of DNMTi is built upon the existing tumor-recognition capacities of $\gamma\delta$ T cells'. One possibility is that NKG2DL (eg MICA/B, ULBPs) are involved. Experiments (eg blocking antibody assays) that assessed the contribution of the $\gamma\delta$ TCR or NKRs such as NKG2D would provide important information on this point.

c) The mechanism invoked to explain DAC-based enhancement of $\gamma\delta$ T cell recognition (ICAM-1 induction, increases in immune synapse stability) does not appear to be very specific either to V δ 1 T cells, or even to the $\gamma\delta$ T cell compartment, as LFA-1 is widely expressed on $\alpha\beta$ T cells. Can DNMTi drugs such as DAC enhance V γ 9V δ 2 T cell recognition? Do DNMTi drugs enhance $\alpha\beta$ TCR/MHC reactivity in a similar ICAM-1-dependent fashion? These questions should either be experimentally address or at least discussed. DAC-induced effects on $\alpha\beta$ T cell responses might have clinical implications, both for induction of anti-tumour responses or autoimmunity.

4. Lung cancer focus and correlations

a) It is unclear why the authors chose to focus on lung cancer, rather than say other malignancies where DNMTi drugs are clinically approved. They should outline this justification more clearly.

b) Arguably the weakest aspect of the study in my view is the cytoskeletal gene signature correlations with survival in lung cancer patients. Whilst I don't doubt these signatures correlate

with survival, it is clear that such cytoskeletal genes will be of core importance to a diverse range of cell types, including but not restricted to immune synapse formation. The analyses don't discriminate between cancer cells, stromal cells, and immune cells, so concluding the effects relate to expression on cancer cells or immune cells, or anti-tumour immunity, is already an assumption; in principle they could relate eg to suppression of metastasis. It is also dangerously naïve to interpret them as tightly linked to $\gamma\delta$ T cell biology (as implied at the end of the Results section), even when TRDC and TRDG genes are assessed alongside. The Discussion appears more nuanced in relation to these points, and proposes the more likely explanation that they reflect a general 'immune responsive' phenotype, but this is not addressed in bioinformatic analyses. A second, related point is that in reality, the frequency of the intratumoural $\gamma\delta$ T cell response in lung cancer is likely to be dwarfed by that driven by the $\alpha\beta$ T cell compartment, and it would be naïve not to consider whether enrichment of $\gamma\delta$ T cells in some patients merely correlates with other, potentially more important aspects of anti-tumour immunity, such as Th1-driven and cytotoxic $\alpha\beta$ T cell responses, particularly given the highly coordinated regulation of intratumoural immunity. These ideas could and should be addressed bioinformatically and I think the somewhat naïve tone of the final Results section should be altered to match that of the Discussion in relation to these points.

Technical criticisms/improvements

- (i) Figure 1e shows upregulation of BTN2A1 in DAC-treated cells. This observation should be discussed. Similarly, on page 11, in relation to $\gamma\delta$ TCR ligands (Supplementary Figure 4c), the authors should also assess BTN2A1 (Rigau et al, 2020; Karunakaran et al, 2020), which has been shown to be critical alongside BTN3A1 for V γ 9V δ 2 T cell recognition.
- (ii) In Figure 5e, 'M' (?mock) and 'D' (? DAC-treated) should be defined explicitly.
- (iii) In the Discussion, the explanation for the discrepancy of the results in the current study and those in reference 60, in relation to suggested effects of epigenetic upregulation of NKG2D ligand or $\gamma\delta$ TCR ligands in promoting V γ 9V δ 2 cytotoxic responses to osteosarcoma cells, could be clarified. Does this reflect a difference in the response of the specific cancer cells studied to the drug, or alternatively differences in V γ 9V δ 2 versus V δ 1 T cell recognition ?
- (iv) In Figure 6b, the word Granzyme is mis-spelled as Ganzyme in the figure itself.

Language/text suggestions

The standard of English is excellent and I picked up a single minor typos/grammatical error:
Page 19: second paragraph, sentence starting 'Additionally...'; a 'the' is required before 'surface'.

REVIEWER COMMENTS

Reviewer #1 (gamma delta T cells, tumours) (Remarks to the Author):

In this study, Weng and colleagues propose a new role for DNA methyltransferase inhibitors (DNMTis) in cancer immunotherapy. They have specifically addressed if DNMTis such as decitabine (DAC) can reshape surface proteins of cancer cells to facilitate MHC-unrestricted recognition by gamma-delta (gd) T cells. They have chosen a cutting-edge protocol that enables the selective enrichment and enhanced antitumor functions of the Vd1 subset of human gd T cells, Delta One T (DOT) cells. They show that DOT cells are significantly more efficient against lung cancer cells when these are pre-treated with DAC, which orchestrates cytoskeleton rearrangements that stabilize and optimize (in terms of key surface protein expression, both for NK receptors and cellular adhesion, like ICAM-1) the immune synapse between DOT cells and cancer cells. Furthermore, the characterized DAC-associated cytoskeleton signature identifies a subset of lung cancer patients with improved survival, and may be useful for patient selection in future clinical trials. These results are novel and of substantial interest to the community.

The study uses cutting-edge molecular methodologies like SILAC and NGS (mRNA-seq and ATAC-seq); robust cell biology (including microscopy-based) approaches; and well established functional assays for T cells, in vitro and in vivo, that provide solid pre-clinical proof-of-principal for their proposition and its implications for cancer immunotherapy. The comprehensive molecular and cell biology assays performed to understand DAC modulation of the actin cytoskeleton are particularly impressive and worthy of publication in a top-tier journal.

The manuscript is well written and easy to follow. Results are in general discussed properly and the references are appropriate, with one outstanding exception mentioned below.

Major issue:

(Pages 7-8) The authors exaggerate the “improvement” of their expansion/differentiation protocol compared to the published “DOT-cell protocol”, while ignoring the most recent publication by that group (Di Lorenzo et al. Cancer Immunol Res 2019) where they further optimized “DOT cell production”. So the authors need to clearly state what they have changed in comparison to the 2019 version of the DOT-cell protocol, namely by not depleting ab T cells. Critically, the expansion, enrichment and phenotype described here is completely in line with the characterization of DOT cells. Therefore, for the benefit of the readers, I suggest using the terminology DOT cells, instead of gd T cells, straight after introducing the expansion protocol, since there are various therapeutic gd T cell products being explored for cancer immunotherapy (see ref. 3, Sebestyen et al. 2020), and it is important to clarify that the authors are employing DOT cells, especially as these are planned to be tested in upcoming clinical trials.

Response: We sincerely thank the reviewer for the insightful comments. We agree with the reviewer that our expansion protocol shares similarities with the 2019 version of the DOT-cell protocol while differing in that we expand DOT-cells directly from peripheral blood mononuclear cells (PBMCs) without the initial step of $\alpha\beta$ T cell depletion. We stated the differences and cited the research work by Di Lorenzo *et al.* Cancer Immunol Res 2019 in the revised manuscript. For simplicity and clarity, we also revised our manuscript to use the terminology DOT-like or DOT cells.

Minor issues:

1. Summary: when referring to a “clinical-grade protocol to enrich for the Vd1 subset”, the authors should add its known designation and also mention its enhancement of effector functions, thus changing to: “clinical-grade protocol to enrich for the Vd1 subset (Delta One T cells) while enhancing their antitumor effector functions”. This should also feature in the title on page 7, which should be modified to: “Ex vivo expansion of Vg1-enriched gd T cells with enhanced antitumor effector functions”.

Response: We agreed with the reviewer and modified the sentences accordingly.

2. The way the experiment in Fig 1 is set up is not very clear, from the Mock: DAC 1:1 mixture (Fig 1a) used for SILAC, to the analysis (Fig. 1b-h) focused on comparisons at D3 and D3R3 (after 3 days after drug removal). Can the authors clarify this better in the text?

Response: We appreciate the reviewer’s constructive comments and clarified the experimental set-up in the revised text as follows.

“Lung cancer cells were differentially labeled by growing in cell culture media containing heavy arginine (L- Arginine- $^{13}\text{C}_6$)/lysine (L-Lysine- $^{13}\text{C}_6$,) or normal arginine/lysine. The SILAC-labeled cells (heavy) and the unlabeled cells (light) were treated with phosphate-buffered saline and 100 nM DAC, respectively, for three consecutive days (D3) and grown in a drug-free medium for another three days (D3R3) (**Fig. 1b**). After biotinylation of the cell surface proteins, total cell lysates of DAC-treated cells at D3 and D3R3 were 1:1 mixed with their corresponding mock-treated counterparts and subjected to surface protein isolation, followed by quantitative proteomics.” “Among all the identified proteins, we focused on 314 proteins that showed sustained upregulation upon DAC treatment by at least 1.4-fold for both D3 and D3R3, for their continued induction after drug withdrawal suggested a possible epigenetic regulation (**Fig. 1d**).”

We modified Figure 1b to better illustrate the timeline of drug treatment and sample collection for DAC- and Mock-treated cells.

In addition, the parameters used in the proteomic analysis were described in detail in Materials and Methods for a better flow of the main text.

3. When mentioning NKG2D ligands and their roles on gd T cell activation (page 6), a more specific review on the topic should be cited, such as Silva-Santos, Strid, *Frontiers Immunol* 2018; or Simões, Di Lorenzo, *Frontiers Immunol* 2018.

Response: We thank the reviewer for the thoughtful reminder. We have cited these important reviews on NKG2D ligands and $\gamma\delta$ T activation in the revised manuscript (page 6).

4. (Page 11) When stating “putative ligands for $\gamma\delta$ TCR, including CD1d complexes³⁰, BTN3A³¹, BTNL3³² and others³³, are not consistently upregulated by DAC”, the authors should qualify that such ligands, maybe with the exception of CD1d for a very minor subset, are NOT applicable to Vd1 T cells.

Response: We agree with the reviewers that some of these $\gamma\delta$ TCR ligands are not applicable to V δ 1 T cells, which is consistent with our point that these ligands are not the determinants for the DAC-potentiating cytotoxicity observed in our study. In this regard, we revised the sentence to prevent potential confusion as follows. “*It is worth noting that other immune synaptic proteins, such as putative ligands for $\gamma\delta$ TCRs on non-V δ 1 cells (e.g., V δ 2 cells), including BTN2A1, BTN3A³³, BTNL3³⁴, and others³⁵, are not consistently upregulated by DAC in all the cancer cell lines from our surface proteomics study (Supplementary Fig. 8a in the revised manuscript), and may not be essential for DOT cell cytolytic activity.*”

5. Figure 6 is of limited interest (only to highly specialized readers) and should be moved to supplementary material.

Response: We adopted the reviewer’s suggestion and moved Figure 6 to Supplementary Fig 13. We also wanted to point out that the information of whether and how DNA demethylating agents affect the immunophenotype and functionality of *ex vivo* expanded $\gamma\delta$ T cells is important for therapeutic purposes.

6. By contrast, Figure 7 is very relevant but has 2 very distinct messages and sets of data, that should be split into 2 main figures (even if each of them small, which

should not be an issue): one with the mouse xenograft model and another with the lung cancer patient stratification.

Response: We appreciate the reviewer's suggestions and split the original Figure 7 into two figures (Figure 6 and Figure 7) in the revised manuscript.

7. The authors do not show DOT-cell infiltration (into tumor lesions) nor persistence in the blood in their xenograft model (Fig 7a-c). Could they provide (or comment on) such data?

Response: In response to the reviewer's request, we performed new mouse experiments for *in vivo* imaging of DOT cell trafficking and homing. We treated NSG mice bearing lung cancer xenografts with DAC and $\gamma\delta$ T cells prestained by CYTO-ID® Red long-term cell tracer dye. The images were taken at 2 and 4 hours after $\gamma\delta$ T injection on day 12 using IVIS® Spectrum. The data showed enhanced homing ability and significant intratumoral infiltration of $\gamma\delta$ T cells in the DAC treatment group. On the other hand, poor tumor infiltration of $\gamma\delta$ T cells was noted in the control group (**Figure 1**). Immunofluorescence staining also revealed increased infiltration of human DOT cells in lung cancer xenograft tumors in the DAC treatment group (**Figure 2**). The data support the conclusion that DAC enhances the anti-tumor properties of DOT cells. We include the new data in the revised manuscript as Figure 6d.

Figure 2. Immunofluorescence staining of human $\gamma\delta$ T cells in lung cancer xenograft tumors. Mouse lung cancer xenograft tumors in the mock, DAC, $\gamma\delta$ T and the combination groups were harvested 4 hours after $\gamma\delta$ T injection on day 12 as described in Figure 1. The data show enhanced intratumoral infiltration of human $\gamma\delta$ T cells (stained by human CD45) in the combination group under a high-resolution confocal microscope. DAPI: 4',6-diamidino-2-phenylindole, as a nuclear counterstain. Scale bar: 500 μ m.

8. Besides the cytoskeleton gene signature, can the authors detect any DOT-cell associated markers in the primary lung cancer tissue analyzed in Fig. 7d?

Response: We performed analysis on $\gamma\delta$ TCR-related genes using the RNA-seq data of the primary lung cancer tissues (NTUH and TCGA cohorts) in Fig. 7. We were able to detect $\gamma\delta$ TCR genes in the NTUH cohort (**Figure 3**). Notably, higher expressions of $\gamma\delta$ TCR genes were detected in the tumor tissues of the immune-sensitive and immune-intermediate groups, which were associated with better overall survivals. On the other hand, low expressions of $\gamma\delta$ TCR genes were observed in the tumor tissues of the immune-insensitive group, which had the worst overall survivals (**Figure 3**). We were not able to detect $\gamma\delta$ TCR-related genes in the TCGA RNA-seq data, possibly due to relatively lower sequencing depth (NTUH, approximately 50 million reads on average; TCGA, approximately 16 million reads on average). Since $\gamma\delta$ T cells usually represent a very small subset of tumor-infiltrating lymphocytes, a sufficient sequencing depth may be required to detect low-abundance transcripts. The $\gamma\delta$ TCR heatmap data were added in the revised manuscript as Supplementary Fig. 14.

Figure 3. Detection of $\gamma\delta$ TCR gene expressions in primary lung adenocarcinoma tumor tissues. A heatmap of $\gamma\delta$ TCR-related genes (i.e., gamma chain and delta chain) derived from mRNA-seq data of primary lung adenocarcinoma tumor tissues in patients at National Taiwan University Hospital (NTUH). The data show that higher levels of $\gamma\delta$ TCR gene expression are

detected in the immune-sensitive/immune-intermediate groups as compared with low $\gamma\delta$ TCR gene expression in the immune-insensitive group.

9. The discussion has some inaccuracies that should be corrected (pages 17-18):
➤ Remove intraepithelial from Vδ1+ T cells, since they also circulate (that's why the authors could expand them from blood!); the following sentence already highlights their preferential tissue localization;

Response: We agree with the reviewer that some Vδ1+ T cells can also be found in the circulation. We removed the term “intraepithelial” for Vδ1+ on page 17 in the revised manuscript.

➤ Modify “the clinical use of Vδ1+ T cells has been fairly limited” to “the clinical use of Vδ1+ T cells has been limited by the lack of a reliable expansion protocol” and mention here the recent advances provided by the DOT-cell protocol (up to its 2019 version, see above), clearly stating that it was the basis for the current study (with minor modifications as clarified above, see major issue).

Response: We modified the sentences according to the reviewer's suggestion. The articles describing the DOT-cell protocols were also cited here. The revised paragraph reads as follows,

“Nevertheless, the clinical use of Vδ1+ T cells has been limited by the lack of a reliable expansion protocol. Instead, Vδ2+ T cells, which can be easily expanded with amino bisphosphonates (e.g., zoledronic acid) or phosphoantigens, are used in most clinical trials with limited clinical efficacy against solid tumors^{5,6}. Moreover, certain γδ T subsets may possess protumor functions⁷; therefore, caution should be taken during the process of cell expansion for therapeutic uses. In light of recent advances in the protocol for DOT cell generation²⁶⁻²⁸, our modified expansion protocol eliminates the need for αβ T depletion and enriches Vδ1+ T cells with antitumor immunity instead of the protumor IL-17-producing γδ T subsets that have been shown to promote tumor progression^{55,56}”

Reviewer #2 (IEL, cancer) (Remarks to the Author):

The present manuscript by Rueyhung R. Weng et al., entitled Epigenetic modulation of immune synaptic-cytoskeletal networks potentiates γδ T cell-mediated cytotoxicity in lung cancer, provides new important information about therapeutic potential of human gd T cells against tumor cells treated with epigenetic modulation of DNA methyltransferase inhibitors. It is a well-designed and performed study and, overall, the manuscript is well written and organized in a systematic manner with solid and convincing experimental results. However, quite a few issues need further clarification and/or experimental data. The major concern stands from the authors statement that "ex vivo expanded human Vδ1-enriched γδ T cells retain antitumor effector functions". This indeed suggest "a Vd1 T cell-dependent anti-tumor

response against DAC-treated cancer cells". Authors should better state and rephrase

Response: We thank the reviewer for stating that "It is a well-designed and performed study and, overall, the manuscript is well written and organized in a systematic manner with solid and convincing experimental results." Indeed, as the reviewer stated, the main focus of this paper is to demonstrate the therapeutic applications of *ex vivo* expanded $\gamma\delta$ T cells against cancer cells pretreated with DNA methyltransferase inhibitors. We would like to echo reviewer#1's comment and point out that the expansion protocol was similar to the protocol for Delta One T (DOT) cell generation with some modifications. It appears that the major cell subset of the resulting cell preparation is V δ 1 cells. It was not our intention to indicate that V δ 1 cells are the only cell subset that exerts anti-tumor responses. We adopted the suggestion by reviewer#1 and used the terminology DOT-like or DOT cells to describe our *ex vivo* expanded cell preparation.

Major point:

1. As shown in Fig. 2a and Supplementary Fig 2c, enrichment of gd T cells accounts about 80 % of total CD3 T cells, in addition Vd1 T cells accounts approximately 70% of all $\gamma\delta$ T cells. Thus, it is not clear whether apoptotic and cytotoxic effects shown in Fig. 2 are due to the specific Vd1 or either Vd2 T cell subset. Moreover, other cells such as CD8 T cells could contribute to the killing of tumor cells. Thus, to directly assess gd T cell-mediated and in particular the specific Vd1-dependent cytotoxic response the authors should set up experiments that would discriminate cytotoxic response and cytokine (TNF- α , IFN- γ) production of different Vd1, Vd2 and CD8 T cell populations (e.g. CD107a flow cytometry-based assay).

Response: We appreciate the reviewer's valuable comments. We agree with the reviewer that, while V δ 1 cells are the major cell population after *ex vivo* expansion, the cell preparation may contain minor subsets of V δ 2 cells and CD8 T cells. To dissect the potential contributions of each cell subpopulation to the overall cytotoxic response, we performed cocultures of H1299 lung cancer cells with sorted V δ 1, V δ 2, and CD8 T cells, respectively. We found that V δ 1 cells displayed relatively higher cytotoxicity than V δ 2 and CD8 T cells (**Figure 4**). Consistently, expressions of CD107a, a functional marker for T cell activation and degranulation, were also higher in V δ 1 cells than V δ 2 or CD8 T cells after coculture with H1299 lung cancer cells (**Figure 5, left panel**). A similar finding was observed in a human ovarian carcinoma cell line, OVCAR-8 (**Figure 5, right panel**), which suggested that this phenomenon was not restricted to a single cancer type. Poor cytolytic activity of CD8 T cells in this allogeneic setting was anticipated due to the MHC/TCR mismatch. The data are included in the revised manuscript as **Supplementary Fig. 7**.

2. Similarly, in Fig. 4 showing immunofluorescence staining of synapse formation the author should rather use FACS-sorted Vδ1 or Vδ2 T cells and not Vδ1-enriched cells.

Response: As the reviewer suggested, we quantified immune synapses formed between lung cancer cells and sorted Vδ1, Vδ2, or CD8 T cells (**Figure 6**) in coculture at an E: T ratio of 3:1. We observed a greater number of immune synapses formed with Vδ1 cells than with Vδ2 cell or CD8 T cells. Consistent with our prior observations, DAC pretreatment markedly enhances immune synapse formation for both Vδ1 and Vδ2 cells. On the other hand, few immune synapses between CD8 T cells and cancer cells were formed since CD8 T cells were isolated from a noncognate donor. The data are included in the revised manuscript as **Supplementary Fig. 7**.

Minor points:

1. No statistically significant expansion of total gd T cells and of the specific Vδ1 T cell-enrichment is shown in Fig. 2a.

Response: We thank the reviewer for the kind reminder. We added statistics in Fig. 2a. One-way ANOVA with Tukey's multiple comparisons test was used to compare expansion folds at different time points (0, 14 and 21 days) for total γδ T cells and for Vδ1 cells.

2. Some figures (e.g. Fig. 2a or Fig. 5e) lack numbers of biological/experimental replicates.

Response: The data in Figure 2a were summarized from three independent expansion experiments. We have added the information in the figure legends. Genome-wide methylation data in figure 5e were obtained from 4 different lung cancer cell lines with and without decitabine (DAC) treatment. Although each cell line has only one set of genomic data (methylome, RNA-seq, ATAC-seq) for each

condition (untreated vs. treated) due to funding limitations during the COVID-19 pandemic, we were able to observe consistent findings across different cell lines.

Reviewer #3 (gamma delta T cells) (Remarks to the Author):

Weng et al present a study that focusses on pharmacological manipulation of $\gamma\delta$ T cell reactivity. This is of significant interest, not least because of the increasing focus on developing $\gamma\delta$ T cell-based immunotherapies, but has largely not been addressed previously outside of drugs (such as aminobisphosphonates) that influence phosphoantigen-mediated activation of V γ 9V δ 2 T cells. The study instead focusses on DNMTI drugs, some of which have been approved for use in certain haematological malignancies (AML, MDS). While their clinical impact is variable between patients and their mode of action is unclear, they have been suggested to exert a range of immunomodulatory effects.

Focussing largely on lung cancer, Weng and colleagues extend this body of knowledge by providing evidence that DNMTI treatment upregulates molecules involved in $\gamma\delta$ T cell activation on cancer cells, increases synapse formation between $\gamma\delta$ T cells and cancer cell targets, and potentiates $\gamma\delta$ T cell anti-tumour immunity, specifically involving V δ 1 rather than V γ 9V δ 2 T cells. Finally, they also suggest that lung cancer patients can be stratified by immune cytoskeleton expression signatures that relate to $\gamma\delta$ T cell-mediated cytotoxicity.

Strong points

The study has several strong points. The identification of drugs that boost V δ 1 tumour recognition (extending pharmacological manipulability beyond the well-established aminobisphosphonate/V γ 9V δ 2 axis), and that can be combined with V δ 1 T cell adoptive transfer, is as far as I am aware novel. In doing so the study brings to light a novel immunomodulatory aspect of DNMTi drugs, although the authors note that others have previously reported DNMTi enhancement of V γ 9V δ 2 cytotoxic responses to osteosarcoma cells (Wang et al, 2018). The current study also nicely applies a range of complementary techniques (SILAC, mass cytometry, microscopy, mouse models) to shed light on some mechanistic aspects of this enhanced V δ 1 tumour targeting. Of note, the importance of ICAM-1 in $\gamma\delta$ T cell recognition of target cells is not novel, and has been previously highlighted, both in the context of V δ 2-negative $\gamma\delta$ T cell recognition of infected and cancerous cells (eg Willcox, C. R. et al. Cytomegalovirus and tumor stress surveillance by binding of a human $\gamma\delta$ T cell antigen receptor to endothelial protein C receptor. *Nat. Immunol.* 13, 872–879 (2012).) and also discussed in reviews (eg Willcox CR and Willcox BE, *Nat Immunol*, 2019). However, the finding that this $\gamma\delta$ T cell/ICAM-1 axis is pharmacologically manipulable is novel. In light of the growing interest in $\gamma\delta$ T cell immunotherapies against cancer, my feeling is the study's findings will be of substantial interest to the field.

Criticisms

1. In some effector assays (eg Figure 2e, f) the impact of combining DAC-treatment of cancer cells and $\gamma\delta$ T cells is relatively weak, and appears to be largely additive (similar to or slightly more than the sum of the separate effects of DAC and $\gamma\delta$ T cells); moreover in at least one case the effect of combination even appears sub-additive (eg Figure 2f). However, if DAC treatment was truly substantially boosting $\gamma\delta$ T cell recognition surely one would expect a synergistic effect? In contrast, in the animal model (eg Figure 7a, b) the effect of combining both DAC and $\gamma\delta$ T cells does appear to be synergistic. This ‘additive effect’ vs ‘synergism’ issue should be directly discussed – including in relation to these different experiments, and with reference to likely underlying mechanisms.

Response: We agree with the reviewer that there are some variabilities in the combinatorial effect of DAC and $\gamma\delta$ T cells between *in vitro* cell assays and *in vivo* mouse experiments. It is possible that, in the mouse experiments, the enhanced combinatorial effect of DAC and $\gamma\delta$ T cells could be partially attributed to the intact architectures of tumor microenvironment with additional cell types and immune mediators to facilitate cell killing, which are missing in the cell-based experiments *in vitro*. In addition, we are aware that, for evaluation of drug-drug interaction, certain quantitative pharmacodynamic models such as Chou-Talalay’s combination index theorem may be required to define synergism. Nevertheless, all quantitative analytic methods for assessing dose-effect relationships are designed for drugs instead of cell therapies. The two types of treatments differ fundamentally in their mechanisms of action and in pharmacodynamics. The existing quantitative methods may not be readily applied to evaluating the combination of a drug and a cell-based treatment. To avoid overgeneralization, we are cautious about using the term “synergy” to describe the phenomenon we observed and used “potentiation” or “enhancement” instead (we realize these terms are also not perfect but closer to reality). What we observed was that DAC appeared to enhance the stress signals of cells and to make more cells susceptible to $\gamma\delta$ T killing. On the other hand, few known mechanisms suggest that $\gamma\delta$ T cells may enhance the cytotoxicity of DAC. In addition, inherent variabilities of different cell lines also add complexity to the observed combined effects, just as we do not anticipate the combination therapy works for all lung cancer patients. Therefore, the combination effects may not simply be classified as “additive” or synergistic.”

2. Regarding the V δ 1-focussed $\gamma\delta$ expansion protocol, I have two queries.

a) Aside from the obvious strong similarities to the expansion procedures developed by Silva-Santos and colleagues (which could be acknowledged more clearly), a second question is the extent to which the procedures as outlined preferentially enrich for V δ 1 T cells. In Figure 2a (and Supplementary Figure 2a), when the V δ 1-focussed expansion protocol is highlighted, the $\gamma\delta$ T cell compartment of the

individual(s) at day 0 appears to atypical in that it is dominated already by V δ 1 T cells. This is quite unusual, as in most individuals V γ 9V δ 2 T cells predominate, and also the resulting V δ 1-dominated post-expansion proportions of $\gamma\delta$ subtypes only reflect a slight enrichment of V δ 1 T cells compared to the starting profile. It would be more informative to show expansion from an individual with a more typical V δ 2-dominated $\gamma\delta$ T cell compartment, particularly as the text highlights the procedure's potential to elicit preferential V δ 1 enrichment. Based on 'normal' relative proportions of V δ 2 vs V δ 1 T cells at day 0 (usually around 4 or 5 to 1), if the same modest levels of V δ 1 enrichment were observed, it is quite likely that the $\gamma\delta$ T cell profile of most individuals following the three-week expansion procedure would actually be dominated numerically by V δ 2 T cells. The authors should therefore show expansions in a range of donors, including those with more normal ratios of V δ 2 to V δ 1 T cells.

Response: We thank the reviewer for the constructive critique. We agree with the reviewer that our expansion protocol shares similarities with the DOT-cell protocol developed by Silva-Santos and colleagues. One of the major differences is that we expand DOT-like cells directly from the PBMCs without the initial step of $\alpha\beta$ T cell depletion. We stated the differences and acknowledged the work by Silva-Santos and colleagues more clearly in the revised manuscript. (See our response to reviewer#1) Moreover, to demonstrate that our expansion protocol is preferentially enriched for V δ 1 T cells, we present the data of another five donors using the current expansion protocol in **Figure 8** (Supplementary Fig.2 in the revised manuscript). As the reviewer stated, in all of the individuals, V δ 2 T cells predominate (around 60 to 80%) before the expansion (Day 0). After *ex vivo* expansion, we observed a significant enrichment of V δ 1 cells (mostly around 80% to 90%) in the resulting cell preparation (Day 21). Consistently, we observed a 1,571 to 14,245-fold expansion of V δ 1 cells (Day 21/Day 0), but only 3 to 59-fold for V δ 2 T cells (**Figure 8**).

b) Particularly in light of the point above (in a), which suggests that mixed populations of different $\gamma\delta$ T cell subsets have been used, the results are weakened significantly by not resolving in many assays which $\gamma\delta$ T cell subsets the readouts

relate to. This is relevant to in vitro cytotoxicity results in Figures 2 and 3, synapse formation in Figures 4 and 5, and even the in vivo data in Figure 7. Given potentially overlapping effector functions of different $\gamma\delta$ T cell subsets (eg cytotoxicity and cytokine production), and the desire to obtain mechanistic clarity, this is suboptimal, and means it is not possible to attribute drug-induced effects to a specific $\gamma\delta$ T cell subset or collectively to the entire $\gamma\delta$ T cell compartment. The authors should include at least some assays where they can unequivocally resolve which subset the assay readouts relate to.

Response: Per reviewers#2 and #3's request, we performed cocultures of H1299 lung cancer cells with sorted V δ 1, V δ 2, and CD8 T cells from our *ex vivo* expanded cell preparations, respectively. We observed that the sorted V δ 1 cell subset demonstrated the highest cytotoxicity against lung cancer cells among the three cell populations. V δ 1 also expressed higher CD107a and formed a more significant number of immune synapses than V δ 2 cells in coculture with lung cancer cells. (Figures. 4, 5, and 6. Response to reviewer#2) (Please refer to Supplemental Fig. 7 in the revised manuscript). These data indicate that V δ 1 cells not only predominant in number but also possess stronger effector functions in our *ex vivo*-expanded cell preparations.

3. The increase in tumour-targeted $\gamma\delta$ T cell reactivity appears to be entirely dependent upon and largely involving ICAM-1 upregulation. Three key considerations follow from this.

a) The specificity of the approach for targeting of tumour cells versus normal cells is unclear. Potentially, ICAM-1 levels could comprise one component of a stress signature sensed by V δ 2-negative T cells. However, one concern would be that use of DNMTi drugs such as DAC may induce $\gamma\delta$ reactivity to non-cancer cells by upregulating ICAM-1 and strengthening synapse formation. This is not addressed as almost all analyses relate to tumour cell recognition, however could ultimately result in off-tumour toxicity. Consistent with this, clinical use of DNMTi drugs (eg in AML) can result in various side effects – it is unclear if these are immune-mediated. The authors comment in the Discussion that no toxicity was observed in the mouse model, but this is a xenograft model where many potentiating receptor/ligand interactions between V δ 1 T cells and mouse target cells will be mismatched. In my opinion the study would therefore be strengthened by any in vitro experiments on non-transformed human cells that address this angle.

Response: We appreciate the concern of off-target effects raised by reviewer#3 since ICAM-1 is a membrane protein that may also be expressed on normal cells and upregulated by DAC. Thus, we performed $\gamma\delta$ T killing assays on the peripheral blood mononuclear cells (PBMCs) known to express ICAM-1 on the cell surface. We observed no significant increase in cytotoxicity of PBMCs upon combination treatment of DAC and *ex vivo* expanded V δ 1-enriched $\gamma\delta$ T cells (DOT-like

cells)(**Figure 9**)(Supplemental Fig. 4e in the revised manuscript). We reasoned that, while increased adhesion molecules may strengthen cell-cell contact and facilitate T cell immunosurveillance, to complete an intricate cytolytic process, other stress signals are also required in addition to the structural components of immune synapses.

b) Related to (a), it is unclear whether ICAM-1 induction on tumour cells over-rides signals other immune receptors often thought critical to $\gamma\delta$ T cell recognition, such as the $\gamma\delta$ TCR or NKRs, or whether such pathways are still critical but crucially potentiated by increased ICAM-1 on target cells. The authors hint at this in the Discussion when they highlight ‘the potentiating effect of DNMTi is built upon the existing tumor-recognition capacities of $\gamma\delta$ T cells’. One possibility is that NKG2DL (eg MICA/B, ULBPs) are involved. Experiments (eg blocking antibody assays) that assessed the contribution of the $\gamma\delta$ TCR or NKRs such as NKG2D would provide important information on this point.

Response: We thank the reviewer for the insightful comments. As shown in **Figure 10** (Supplementary Fig. 8b in the revised manuscript), we performed $\gamma\delta$ TCR and NKG2D blocking assays and found that blocking of NKG2D attenuated $\gamma\delta$ T-mediated cytotoxicity of DAC-pretreated lung cancer cells. This suggested that the NKG2D receptor-ligand axis is important in immune recognition and tumor lysis by $\gamma\delta$ T cells. The induction of ICAM-1 by DNMTi does not completely override the conventional tumor-recognition signals between cancer and $\gamma\delta$ T cells.

c) The mechanism invoked to explain DAC-based enhancement of $\gamma\delta$ T cell recognition (ICAM-1 induction, increases in immune synapse stability) does not appear to be very specific either to $V\delta 1$ T cells, or even to the $\gamma\delta$ T cell compartment, as LFA-1 is widely expressed on $\alpha\beta$ T cells. Can DNMTi drugs such as DAC enhance $V\gamma 9V\delta 2$ T cell recognition? Do DNMTi drugs enhance $\alpha\beta$ TCR/MHC reactivity in a similar ICAM-1-dependent fashion? These questions should either be experimentally address or at least discussed. DAC-induced effects on $\alpha\beta$ T cell responses might have clinical implications, both for induction of anti-tumour responses or autoimmunity.

Response: We agree with reviewer#3 that ICAM-1 is a general adhesion molecule and may be recognized by other T cell subtypes, including $V\gamma 9V\delta 2$ T cells and $\alpha\beta$ T cells. As shown above, in the responses to reviewer #2 (**Figure 4 and 6**), we demonstrated that DNMTi may enhance cytolytic effects of sorted $V\delta 1$, $V\delta 2$ and CD8 $\alpha\beta$ T cells on lung cancer cells to various extents (**Figure 4**). Interestingly, while DNMTi enhances immune synapse formation for both $V\delta 1$ and $V\delta 2$ cells (**Figure 6**), $V\delta 1$ cells seemed to demonstrate a greater cytolytic effect with DAC potentiation than $V\delta 2$ cells (**Figure 4**). This indicated that robust cell-cell adhesion is one of many factors required for effective cytolytic immune responses. It is possible that a similar ICAM-1-dependent mechanism may be involved in the interaction between cancer and $\alpha\beta$ T cells in an MHC-restricted manner. The modest potentiation effect of DAC on CD8 T cells observed in Figures 4 and 6 could be attributed to mismatched MHC/TCR molecules. We added the new data in Supplementary Fig. 7 and discussed the data in the Discussion section of the revised manuscript.

4. Lung cancer focus and correlations

a) It is unclear why the authors chose to focus on lung cancer, rather than say other malignancies where DNMTi drugs are clinically approved. They should outline this justification more clearly.

Response: DNMTis have been approved by the FDA for the treatment of hematological malignancies. Nevertheless, therapeutic efficacies of DNMTi in solid tumors are less satisfactory when used alone. Thus, much effort has focused on

investigating the role of DNMTs as a sensitization/priming agent in combination with other treatment modalities, rather than a merely cytotoxic agent, for the management of solid tumors. The key is to provoke epigenetic changes on the cellular or molecular levels to enhance cancer cells' susceptibility to other treatments such as immunotherapy, chemotherapy, or radiotherapy. In these settings, DNMTs are often used at low doses with little direct cytotoxicity or fewer side effects (Chiappinelli *et al.* Cell 2015; Agrawal *et al.* Pharmacol Ther 2018). Lung cancer is the leading cause of cancer death worldwide despite rapid advances in therapeutic strategies. Our team is composed of clinicians and scientists who have expertise in epigenetics, lung cancer biology as well as T cell-based immunotherapy. It is our goal to develop new therapeutic strategies to help lung cancer patients who do not respond to existing treatment options. Moreover, we believe our findings of epigenetically-potentiated $\gamma\delta$ T cell therapy have potential implications in other hematological and epithelial cancers as well. As shown in the manuscript, we demonstrated similar effects of DAC-induced cytoskeleton-remodeling and $\gamma\delta$ T killing in colon cancer cells (Fig. 5 and Supplementary Fig. 4). We hope that this proof-of-principle study opens up new possibilities for the treatment of not only lung but also other cancer types to help patients in need. We briefly outlined the justification in the Introduction section.

b) Arguably the weakest aspect of the study in my view is the cytoskeletal gene signature correlations with survival in lung cancer patients. Whilst I don't doubt these signatures correlate with survival, it is clear that such cytoskeletal genes will be of core importance to a diverse range of cell types, including but not restricted to immune synapse formation. The analyses don't discriminate between cancer cells, stromal cells, and immune cells, so concluding the effects relate to expression on cancer cells or immune cells, or anti-tumour immunity, is already an assumption; in principle they could relate eg to suppression of metastasis. It is also dangerously naïve to interpret them as tightly linked to $\gamma\delta$ T cell biology (as implied at the end of the Results section), even when TRDC and TRDG genes are assessed alongside. The Discussion appears more nuanced in relation to these points, and proposes the more likely explanation that they reflect a general 'immune responsive' phenotype, but this is not addressed in bioinformatic analyses. A second, related point is that in reality, the frequency of the intratumoural $\gamma\delta$ T cell response in lung cancer is likely to be dwarfed by that driven by the $\alpha\beta$ T cell compartment, and it would be naïve not to consider whether enrichment of $\gamma\delta$ T cells in some patients merely correlates with other, potentially more important aspects of anti-tumour immunity, such as Th1-driven and cytotoxic $\alpha\beta$ T cell responses, particularly given the highly coordinated regulation of intratumoural immunity. These ideas could and should be addressed bioinformatically and I think the somewhat naïve tone of the final Results section should be altered to match that of the Discussion in relation to these points.

Response: We thank the reviewer for the thoughtful comments. We agree with the reviewer that the cytoskeletal signature may reflect a much more complex cancer-immune interaction in general rather than indicating an exclusive association with $\gamma\delta$ T cell biology. This cytoskeletal signature, derived from DAC-treated lung cancer cell lines, may represent an intrinsic feature of cancer cells for immune reactivity assessment and can potentially be used for patient stratification for responses to cell-based immunotherapy in the clinical setting. As shown above, in response to reviewer#1 (**Figure 3** above; Supplementary Fig. 14 in the revised manuscript), we corroborated the association between this cytoskeletal signature and $\gamma\delta$ T cells by revealing higher expressions of general $\gamma\delta$ TCR genes in the tumor tissues of the immune-sensitive and immune-intermediate groups. To further demonstrate that our cytoskeletal signature reflects a general “immune responsive” phenotype, we performed a computational deconvolution of transcriptomic data from the TCGA lung cancer tissues using a CIBERSORT-LM7 reference gene signature matrix, generated by a modified CYBERSORT algorithm proposed by Tosolini *et al.* Oncoimmunology 2017. The LM7 reference gene matrix enabled computational identification of $\gamma\delta$ T cells (mostly V γ 9V δ 2 because few V δ 1 transcriptomic datasets were available) that could not be recognized by the original CIBERSORT algorithm (CIBERSORT-LM22). We found that gene signatures for other immune cell types, including CD4 and CD8 T cells, B cells, granulocytes, monocytes, and NK cells, were highly enriched in the immune-sensitive group as compared with those in the immune-intermediate and immune-insensitive groups (**Figure 11**). This suggested that our cytoskeletal signature does correlate with an immune responsive phenotype. We modified the tone of the Results section to match that of the Discussion and incorporated this data in the revised manuscript as Supplementary Fig. 15.

Notably, while our analysis in the TCGA lung cancer tissues showed a high concordance between intratumoral $\gamma\delta$ and $\alpha\beta$ T signatures within individual patient groups stratified by the cytoskeletal signature, the study by Tosolini *et al.* Oncoimmunology 2017 found no correlation between intratumoral $\gamma\delta$ and $\alpha\beta$ T cell abundance across approximately 10,000 cancer biopsies from 50 cancer types using the same deconvolution approach on bulk tumors (i.e., no cytoskeletal signature stratification). Moreover, the abundance of tumor-infiltrating $\gamma\delta$ T cells ($\gamma\delta$ TILs) was variably correlated with patient outcome in the Tosolini study. The data highlight the potential added values of our cytoskeletal signatures in relation to conventional immune cell signatures for patient selection and prognosis prediction.

Technical criticisms/improvements

(i) Figure 1e shows upregulation of BTN2A1 in DAC-treated cells. This observation should be discussed. Similarly, on page 11, in relation to $\gamma\delta$ TCR ligands (Supplementary Figure 4c), the authors should also assess BTN2A1 (Rigau *et al.*, 2020; Karunakaran *et al.*, 2020), which has been shown to be critical alongside BTN3A1 for V γ 9V δ 2 T cell recognition.

Response: We appreciate the reviewer’s insightful and constructive comments. Two seminal papers published earlier this year (Rigau *et al.*, 2020; Karunakaran *et al.*, 2020) identified BTN2A1 as a novel ligand for human V γ 9/V δ 2 T cells. BTN2A1 and BTN3A1 form complexes and elicit V γ 9/V δ 2 T cell response to phosphoantigens. In our study, upregulation of BTN2A1 was observed in the surface proteome of DAC-treated A549 lung cancer cells. Nevertheless, there was no significant enhancement of BTN2A1 protein expression by DAC in the other two cell lines (i.e., CL1-0 and H1299) (**Figure 12**) where DAC potentiation of V δ 1-mediated cytolytic effects took place. Thus, we reason that BTN2A1 may not be the essential molecule for V δ 1 T cell recognition. We added the BTN2A1 data in Supplementary 8a and discussed the observation in the revised manuscript.

(ii) In Figure 5e, 'M' (?mock) and 'D' (? DAC-treated) should be defined explicitly.

Response: We thank the reviewer for pointing this out. We defined “M: mock-treated. D: DAC-treated” in the revised figure legend.

(iii) In the Discussion, the explanation for the discrepancy of the results in the current study and those in reference 60, in relation to suggested effects of epigenetic upregulation of NKG2D ligand or $\gamma\delta$ TCR ligands in promoting V γ 9V δ 2 cytotoxic responses to osteosarcoma cells, could be clarified. Does this reflect a difference in the response of the specific cancer cells studied to the drug, or alternatively differences in V γ 9V δ 2 versus V δ 1 T cell recognition ?

Response: There are several major differences between our study and that by Wang *et al.* First of all, the doses of decitabine used in the osteosarcoma study were in the micromolar range (i.e., 5 to 20 μ M), which are much higher than clinically-achievable serum concentrations in real patients. It has been known that decitabine at high doses *in vitro* may lead to a large number of gene upregulation that may not reflect real molecular changes at physiological doses in the clinical settings. In our study, we chose to use DAC in the nanomolar range (i.e., 100nM), which was considered clinically-relevant low doses based on our previous studies (Tsai *et al.* Cancer Cell 2012). At low doses, we did not observe consistent upregulation of NKG2D ligands (i.e., MICB and ULBP1) across the lung cancer cells we tested. This variability may also be attributed to differences in cancer types since each cancer type harbors both tissue-specific and cancer-specific DNA methylation patterns and may respond differently to DNMTis. On the other hand, the participation of NKG2DL/NKG2D axis in V γ 9V δ 2 or V δ 1 T cell recognition of cancer cells cannot be completely negated as blocking of NKG2D appeared to partially attenuate V δ 1 T cell-mediated killing (**Figure 10** in this letter) although not as significant and dramatic as blocking ICAM-1 interaction.

(iv) In Figure 6b, the word Granzyme is mis-spelled as Ganzyme in the figure itself.

Response: We thank the reviewer for his/her careful reading of the manuscript. We have corrected the typographical error in Figure 6b (now Supplementary Fig. 13b in the revised manuscript per Reviewer#1's request).

Language/text suggestions

The standard of English is excellent and I picked up a single minor typos/grammatical error:

Page 19: second paragraph, sentence starting 'Additionally...'; a 'the' is required before 'surface'.

Response: We thank the reviewer for the kind comments. We corrected our grammatical error and added a "the" before "surface immunome" in the sentence.

REVIEWERS' COMMENTS

Reviewer #1 (Remarks to the Author):

The authors have improved their study through addition of new interesting data and important clarifications in the manuscript. Just a small detail to fix before publication:

- Lines 163-5, the sentence should be modified (SEE CAPS) to: "our group FURTHER optimized a sequential cytokine stimulation protocol (IN THE PRESENCE OF TCR/ CD3 AGONIST ANTIBODIES) for ex vivo clinical-grade expansion of $\gamma\delta$ T cells preferentially enriched for the V δ 1+ subset, "Delta One T" (DOT cells) ADD REF. 26 HERE (...)

Reviewer #2 (Remarks to the Author):

No Further comments

Reviewer #3 (Remarks to the Author):

Comments for the Author

I can confirm that the authors have done an impressive overall job at addressing my comments, and, it seems, the comments of the other reviewers. I feel the changes they have made substantially improve the manuscript, and I feel strongly the study now warrants publication. Key changes are in relation to bolstering data on the expansion protocol itself (response to my point 2a), which cell types are specifically involved (point 2b), relating to potential off-target effects (point 3a), integration of ICAM-1-mediated/DAC-induced effects with other receptor/ligand axes (point 3b), and relevance of the approach to other immune subsets (3c). I also feel that the revised bioinformatics data analyses they include (in response to my point 4) makes for a much more mature assessment of the signatures they highlight, which positions these more realistically within a broad intratumoural immune response.

I am therefore happy to recommend publication, but have three related points that I feel are important to address as the manuscript is progressed to the final version; and some minor corrections.

(i) In regards to point 3a (off-target effects), the authors now include (in Figure 9) killing assays on DAC-treated normal PBMC. This is acceptable but inevitably is a fairly simplistic approximation for the broad spectrum of normal cells that a V δ 1-based immunotherapy could potentially encounter in human patients.

(ii) In response to point 3c (relevance of mechanism to other cell types), the authors now include some data on CD8 T cells (in Figures 4 and 6), showing modest potentiation by DAC. However these assays are based on alloreactive recognition, and could so again is quite a limited exploration of this issue.

(iii) Particularly given the fast pace of development of $\gamma\delta$ T cells towards the clinic, and likelihood of new clinical trials in this space, I feel both of these limitations should be raised in the Discussion, perhaps ideally in a paragraph specifically discussing limitations of the study. Another obvious limitation would be use of immunocompromised xenograft models where effects of DAC-induced ICAM-1 upregulation on T cell tolerance in general would not likely be observed. My recommendation to highlight such limitations is based on the expectation that the results presented in the study may genuinely impact clinically. Doing so could usefully inform groups aiming to devise new clinical trials, where safety considerations will obviously be a major concern.

Minor corrections

1) Finally, Reviewer 1's point (point 4) regarding ligands (that the ligands mentioned by the authors are exclusively for V δ 1-negative cells) is not quite accurate, and as a result the sentence reworded by the authors is also inaccurate. While this is the case for BTN2A1, BTN3A (ie both are

ligands for non-V δ 1 TCRs), this is not the case for BTNL3, which is recognised by gut-localised subsets via $\gamma\delta$ TCRs that are V γ 4+, and predominantly V δ 1+ (see reference 36, Willcox et al, 2019 on BTNL3). A potential rewording (splitting the relevant comment into two sentences) that is accurate is suggested below (NB reference numbers are not changed, elements changed are in bold):

"Consistently, it is worth noting that the putative ligands for $\gamma\delta$ TCRs on non-V δ 1 cells (e.g., V δ 2 cells), including **BTN2A133,34**, **BTN3A35**, and others³⁷, are not consistently upregulated by DAC in all the cancer cell lines from our surface proteomics study (Supplementary Fig. 8a), nor is **BTNL336**, a ligand for V γ 4V δ 1 TCRs. Such ligands therefore may not be essential for $\gamma\delta$ cell cytolytic activity in this context."

Of course it is unclear to me what proportion of V δ 1 T cells used in such assays are V γ 4+, and this may govern the influence of BTNL3 on cytotoxicity in such assays, however this amended comment is still accurate.

2) In the section of the Discussion addressing DAC effects on NKGDL (around line 457), I think it would make sense to briefly mention the finding now included in the revised manuscript that NKG2D/NKG2DL interactions nevertheless likely contribute to $\gamma\delta$ killing of DAC-treated cells.

3) One sentence (lines 459-461) in the Discussion appears slightly overreaching the clinical data at this point. It states:

Thus, our finding of DNMTi-mediated cytoskeletal reorganization via a coordinated epigenetic process provides a missing link behind the effectiveness of $\gamma\delta$ T-based therapy.

This may be the case but I think it would be wise and more justified to rephrase this marginally to be somewhat more cautious and realistic:

"Thus, our finding of DNMTi-mediated cytoskeletal reorganization via a coordinated epigenetic process potentially provides a missing link to enhance the effectiveness of $\gamma\delta$ T-based therapy."

Signed: Professor Benjamin E. Willcox
Institute of Immunology and Immunotherapy, University of Birmingham, UK

REVIEWER COMMENTS

Reviewer #1 (Remarks to the Author):

The authors have improved their study through addition of new interesting data and important clarifications in the manuscript. Just a small detail to fix before publication:
- Lines 163-5, the sentence should be modified (SEE CAPS) to: “our group FURTHER optimized a sequential cytokine stimulation protocol (IN THE PRESENCE OF TCR/ CD3 AGONIST ANTIBODIES) for ex vivo clinical-grade expansion of $\gamma\delta$ T cells preferentially enriched for the V δ 1+ subset, “Delta One T” (DOT cells) ADD REF. 26 HERE (...)

Response: Thank you. We have revised the sentence accordingly.

Reviewer #2 (Remarks to the Author):

No Further comments

Response: Thank you. We are glad that we fully addressed the previous comments by Reviewer#2.

Reviewer #3 (Remarks to the Author):

Comments for the Author

I can confirm that the authors have done an impressive overall job at addressing my comments, and, it seems, the comments of the other reviewers. I feel the changes they have made substantially improve the manuscript, and I feel strongly the study now warrants publication. Key changes are in relation to bolstering data on the expansion protocol itself (response to my point 2a), which cell types are specifically involved (point 2b), relating to potential off-target effects (point 3a), integration of ICAM-1-mediated/DAC-induced effects with other receptor/ligand axes (point 3b), and relevance of the approach to other immune subsets (3c). I also feel that the revised bioinformatics data analyses they include (in response to my point 4) makes for a much more mature assessment of the signatures they highlight, which positions these more realistically within a broad intratumoural immune response.

I am therefore happy to recommend publication, but have three related points that I feel are important to address as the manuscript is progressed to the final version; and some minor corrections.

(i) In regards to point 3a (off-target effects), the authors now include (in Figure 9) killing assays on DAC-treated normal PBMC. This is acceptable but inevitably is a fairly simplistic approximation for the broad spectrum of normal cells that a V δ 1-based immunotherapy could potentially encounter in human patients.

(ii) In response to point 3c (relevance of mechanism to other cell types), the authors now include some data on CD8 T cells (in Figures 4 and 6), showing modest potentiation by DAC. However these assays are based on alloreactive recognition, and could so again is quite a limited exploration of this issue.

(iii) Particularly given the fast pace of development of $\gamma\delta$ T cells towards the clinic, and likelihood of new clinical trials in this space, I feel both of these limitations should be raised in the Discussion, perhaps ideally in a paragraph specifically discussing limitations of the study. Another obvious limitation would be use of immunocompromised xenograft models where effects of DAC-induced ICAM-1 upregulation on T cell tolerance in general would not likely be observed. My recommendation to highlight such limitations is based on the expectation that the results presented in the study may genuinely impact clinically. Doing so could usefully inform groups aiming to devise new clinical trials, where safety considerations will obviously be a major concern.

Response: We really appreciate Prof. Willcox's valuable and constructive comments. We now added a paragraph in the Discussion section regarding the issues raised by Prof. Willcox.

“Likewise, the safety and efficacy of combined treatment of DNMTi and ex vivo expanded V δ 1 cells can be context-dependent. In particular, while we observed little toxicity on normal PBMCs (Supplementary Fig. 4e), the finding does not negate the potential off-target effects on other normal cell or tissue types, which should be taken into consideration when designing clinical trials. In addition, although we demonstrated the potentiating effect of DNMTi on $\gamma\delta$ T-mediated tumor lysis in an immune-compromised mouse model with little interference of other immune cell types, the model is limited in recapitulating the complex interactions between cancer cells and an intact immune microenvironment. Since ICAM-1 is a general adhesion molecule and may be recognized by other T cell subtypes, such as $\alpha\beta$ T cells, it is possible that a similar ICAM-1-dependent mechanism may be involved in the interaction between cancer and $\alpha\beta$ T cells in an MHC-restricted manner. This might potentially enhance the efficacy of immunotherapy by involving more T cells with cytolytic functions. The modest potentiation effect of DAC on CD8 T cells we observed (Supplementary Fig. 7) is probably due to alloreactive recognition. Further investigation on the roles of $\alpha\beta$ T and other immune cell types in DNMTi-potentiated $\gamma\delta$ T therapy is warranted.”

Minor corrections

1) Finally, Reviewer 1's point (point 4) regarding ligands (that the ligands mentioned by

the authors are exclusively for V δ 1-negative cells) is not quite accurate, and as a result the sentence reworded by the authors is also inaccurate. While this is the case for BTN2A1, BTN3A (ie both are ligands for non-V δ 1 TCRs), this is not the case for BTNL3, which is recognised by gut-localised subsets via $\gamma\delta$ TCRs that are V γ 4+, and predominantly V δ 1+ (see reference 36, Willcox et al, 2019 on BTNL3). A potential rewording (splitting the relevant comment into two sentences) that is accurate is suggested below (NB reference numbers are not changed, elements changed are in bold):

“Consistently, it is worth noting that the putative ligands for $\gamma\delta$ TCRs on non-V δ 1 cells (e.g., V δ 2 cells), including BTN2A1^{33,34}, BTN3A³⁵, and others³⁷, are not consistently upregulated by DAC in all the cancer cell lines from our surface proteomics study (Supplementary Fig. 8a), nor is BTNL3³⁶, a ligand for V γ 4V δ 1 TCRs. Such ligands therefore may not be essential for $\gamma\delta$ cell cytolytic activity in this context.”

Of course it is unclear to me what proportion of V δ 1 T cells used in such assays are V γ 4+, and this may govern the influence of BTNL3 on cytotoxicity in such assays, however this amended comment is still accurate.

Response: We agreed with Prof. Willcox’s comments and revised the sentence accordingly.

2) In the section of the Discussion addressing DAC effects on NKGDL (around line 457), I think it would make sense to briefly mention the finding now included in the revised manuscript that NKG2D/NKG2DL interactions nevertheless likely contribute to $\gamma\delta$ killing of DAC-treated cells.

Response: We added a few sentences describing the finding of NKG2D/NKG2DL in the Discussion section as follows,

“We demonstrated that blocking of NKG2D attenuated $\gamma\delta$ T-mediated cytotoxicity of DAC-pretreated H1299 lung cancer cells (Supplementary Fig. 8b), which suggests that conventional tumor recognition mechanisms by $\gamma\delta$ T cells, such as NKG2D/NKG2DL interactions, partially contribute to $\gamma\delta$ T killing of DAC-treated cells.....”

3) One sentence (lines 459-461) in the Discussion appears slightly overreaching the clinical data at this point. It states:

Thus, our finding of DNMTi-mediated cytoskeletal reorganization via a coordinated epigenetic process provides a missing link behind the effectiveness of $\gamma\delta$ T-based therapy.

This may be the case but I think it would be wise and more justified to rephrase this marginally to be somewhat more cautious and realistic:

“Thus, our finding of DNMTi-mediated cytoskeletal reorganization via a coordinated epigenetic process potentially provides a missing link to enhance the effectiveness of $\gamma\delta$ T-based therapy.”

Signed: Professor Benjamin E. Willcox
Institute of Immunology and Immunotherapy, University of Birmingham, UK

Response: We thank Prof. Willcox for his thoughtfulness. We have reproduced the sentence verbatim in our revised manuscript.